# Towards Understanding Evolving Patterns in Sequential Data

**Qiuhao Zeng**
Western University
qzeng53@uwo.ca

**Long-Kai Huang**
Tencent AI Lab
hlongkai@gmail.com

**Qi Chen**
Laval University
qi.chen.1@ulaval.ca

**Charles Ling**[*]
Western University
charles.ling@uwo.ca

**Boyu Wang**[*]
Western University
bwang@csd.uwo.ca

## Abstract

In many machine learning tasks, data is inherently sequential. Most existing algorithms learn from sequential data in an auto-regressive manner, which predicts the next unseen data point based on the observed sequence, implicitly assuming the presence of an *evolving pattern* embedded in the data that can be leveraged. However, identifying and assessing evolving patterns in learning tasks heavily relies on human expertise, and lacks a standardized quantitative measure. In this paper, we show that such a measure enables us to determine the suitability of employing sequential models, measure the temporal order of time series data, and conduct feature/data selections, which can be beneficial to a variety of learning tasks: time-series forecastings, classification tasks with temporal distribution shift, video predictions, etc. Specifically, we introduce the EVOLVING RATE (EVORATE), which quantifies the evolving patterns in the data by approximating mutual information between the next data point and the observed sequence. To address cases where the correspondence between data points at different timestamps is absent, we develop EVORATE$_\mathcal{W}$, a simple and efficient implementation that leverages optimal transport to construct the correspondence and estimate the first-order EVORATE. Experiments on synthetic and real-world datasets including images and tabular data validate the efficacy of our EVORATE method.

## 1 Introduction

Sequential data is ubiquitous across various machine learning tasks, including multivariate time series [33, 38, 44], video streams in computer vision [18, 52, 54], textual data in natural language processing [9, 17, 34], and state-action trajectories in reinforcement learning [5, 45, 56]. Learning with sequential data usually involves predicting future data points, fostering the development of auto-regressive techniques that learn to forecast the subsequent unseen entries in a sequence. Despite the progress in this field, one fundamental challenge persists: the identification of underlying evolving patterns often depends heavily on the subjective interpretations and prior knowledge of human experts. This reliance on subjective judgment lacks a robust quantitative method to assess the evolving patterns over the high-dimensional data in deep learning. For example, when designing a recommendation system, certain products such as clothing are highly dependent on temporal factors (e.g., seasons, fashion trends), while others, like computers, are more influenced by individual customer preferences. Therefore, it is critical to identify and quantify the underlying evolving patterns for different products and integrate this information into the algorithmic design.

---

[*]Corresponding authors: Boyu Wang, Charles X. Ling.

38th Conference on Neural Information Processing Systems (NeurIPS 2024).

Specifically, the following questions are essential but unresolved yet in literature: i) **How can the existence of evolving patterns in data sequences be determined?** Determining the existence of evolving patterns in data is a critical task. It is possible that the data points of a sequence are entirely independent and no evolving patterns exist. For instance, consider the scenario of a person repeatedly tossing a coin. In this case, historical information does not influence the outcome of the next toss. ii) **Can one determine the historical span that significantly influences the current time point?** For example, how do we determine the order (the optimal number of past observations) of an autoregressive model in a principled way? iii) **How can we determine if the collected features are sufficient to reveal evolving patterns?** For instance, to achieve better weather forecasting, how can one determine the essential features, such as altitude, humidity, and geographic location, for gathering a comprehensive set of information for forecasting?

In this work, we address these questions through a unified framework by introducing EVORATE (EVOLVING RATE), a novel approach designed to quantify the evolving patterns of data sequences. EVORATE leverages mutual information as a measure of the existence of the evolving patterns in the data. Notably, while there is a rich history of mutual information estimation in the existing literature [8, 1, 27, 10, 29], existing works ignore the underlying temporal dependency between the data points, and therefore are not well-designed for sequential data. EVORATE tackles this issue by estimating mutual information in an autoregressive manner when learning the compressed embedding from the observed sequence, thereby addressing the aforementioned questions: i) it can serve as an indicator to show that learning a sequential model is not feasible to learn the provided sequential dataset. ii) EVORATE can provide a quantitative measure of the temporal dependency of a sequence, allowing us to control the trade-off between computational complexity and learning performance. iii) EVORATE can also guide us in selecting the most informative features for model training for sequential data.

However, EVORATE is difficult to estimate when dealing with temporal data characterized by snapshots captured at disparate timestamps without clear correspondence between them [30, 48, 42], as we do not track the same data point over different timestamps and thus lack access to its corresponding sample. This scenario hinders the estimation of EvoRate, due to the absence of the correspondence. To mitigate this issue, we propose an enhanced version of our methodology, EVORATE$_\mathcal{W}$, which is specifically designed to establish correspondence among data points across different timestamps utilizing optimal transport within the Wasserstein distance metric, thereby facilitating the estimation of the first-order EVORATE. In all, the benefits of EVORATE to be highlighted include:

- EVORATE enables quantitatively measuring the evolving patterns existing in high-dimensional sequential data by utilizing the neural mutual information estimator. Furthermore, it can be applied to assess temporal order and conduct feature selections in sequential data.
- We further proposed EVORATE$_\mathcal{W}$ to leverage optimal transport to build the correspondence between snapshots at the different timestamps, and hence allow the MI approximations.
- We motivate through analysis the use of mutual information as indicators of evolving patterns and show optimal transport can mitigate the without correspondence issue.
- Synthetic and real-world datasets verify that EVORATE can be a good indicator for evolving patterns, supporting our claim of its benefits. We also design an EDG algorithm based on the insight of EVORATE$_\mathcal{W}$ and verify its performance. The codes are available on GitHub: https://github.com/HardworkingPearl/EvoRate.

## 2 Related Works

**Sequential Data** The analysis and processing of sequential data is driven by diverse applications ranging from video predictions to time series forecasting [25, 50, 9, 16, 33]. Pioneering works such as Long Short-Term Memory (LSTM) [25] networks have established foundational principles for handling long-range dependencies in sequence data. Building on this, the Transformer [50] introduced a revolutionary approach through self-attention mechanisms, enhancing flexibility in handling sequence dependencies. The versatility of Transformers has been further demonstrated in models such as GPT-3 [9] and BERT [16]. Beyond text, sequential data analysis in machine learning also extends to time-series forecasting [33]. Moreover, the application of Graph Neural Networks in capturing dependencies in irregular sequences underscores the breadth of methodologies exploring

the complexities of sequential data [7]. However, a qualitative method for measuring the intensity of evolving patterns remains lacking in the literature.

**Mutual Information (MI) Estimation** has become a pivotal tool in machine learning [39, 8, 1, 27, 10, 29], enabling insights into dependencies that extend beyond traditional correlation measures. In feature selection, MI offers a data-driven approach to identify relevant features without strong assumptions about data distributions [39]. Mutual Information Neural Estimation (MINE) [8] applies deep learning to estimate MI in high-dimensional settings, providing a new methodology for analyzing neural network training dynamics. MI's application in variational inference, especially in the training of variational autoencoders (VAEs) [1]. In reinforcement learning, MI has been used to enhance exploration strategies by quantifying information gain [27]. MI also improves the performance of generative adversarial networks (GANs) [10]. Furthermore, in unsupervised and semi-supervised learning, MI maximization has been shown to effectively leverage unlabeled data [29]. However, none of them employ MI as an indicator for evolving patterns of sequential data.

**Optimal Transport (OT)** has emerged as a powerful framework in machine learning [51, 3, 31, 13, 41], offering a principled approach to compare probability distributions. Optimal transport theory has been leveraged for applications ranging from domain adaptation to generative modeling [51]. Recent advances include the integration of OT with deep learning architectures; Wasserstein GAN (WGAN) utilizes the Wasserstein distance to improve the stability of training GANs [3]. Furthermore, optimal transport has been applied effectively in NLP [31]. The computational aspect of OT has also seen significant developments, Sinkhorn [13] as a scalable method approximates transport plans efficiently. More recently, researchers have explored the differential properties of transport plans in dynamic environments [41]. EVORATE employs OT to recover the correspondence between two consecutive timestamps, facilitating approximations of mutual information.

**Patterns estimation for sequential data** has only one related work in the literature ForeCA [24], which proposes a similar concept, "forecastibility", which measures the uncertainty of the entropy of the spectral density. However, ForeCA has two drawbacks. Firstly, ForeCA can not be used in deep learning as an unacceptable huge computational consumption for real-world high-dimensional data (audio, videos, etc.). In contrast, EVORATE shows the prediction power by relying on mutual information, which tells the ability to predict another variable based on known observed variables. Secondly, while temporal patterns can include trends, cycles, irregular fluctuations, and more complex behaviors, ForeCA can only detect cycled patterns. Instead, EVORATE relies on the neural mutual information estimator, which is known as a good measurement for various patterns as a result of the strong fitting power of neural nets [8, 11, 37, 46].

# 3 Preliminary

## 3.1 Variational mutual information estimation

The mutual information between two random variables $X$ and $Y$ is defined as the KL divergence $D_{\mathrm{KL}}$ between their joint distribution and the product of their marginal distributions:

$$I(X;Y) = D_{\mathrm{KL}}(P(X,Y)||P(X)P(Y)), \tag{1}$$

where we aim to estimate this using samples from $P(X,Y)$; in some cases, the density of the marginals such as $P(X)$ may be known. A wide range of variational methods are designed to estimate variational mutual information [8, 11, 46, 36, 37]. We then use the below estimator to estimate mutual information:

$$\hat{I}(X;Y) := \mathbb{E}_{\hat{P}(X,Y)}[m(x,y)] - \log \mathbb{E}_{\hat{P}(X)\hat{P}(Y)}[e^{m(x,y)}], \tag{2}$$

where $X$ is the random variable, $x$ is a realization of $X$ (as is the case with $Y$ and $y$), $\hat{P}$ is the empirical distribution associated with a dataset of i.i.d. samples, and $m(x,y)$ is a critic function to quantify the similarity between $X$ and $Y$, usually realized by a neural network [8, 11, 46, 36, 37]. We show that MI is highly related to the evolving patterns of the sequential data in Section 4.2.

## 3.2 Optimal transport

A rich class of divergences between probability distributions is induced by the optimal transport (OT) problem [51]. Kantorovich's formulation of the problem is given by

$$W_c(P(X), P(Y)) = \inf_{\pi \in \Pi(P(X), P(Y))} \mathbb{E}_{(X,Y) \sim \pi}[c(X, Y)], \qquad (3)$$

where $c(x, y) : \mathcal{X} \times \mathcal{Y} \to \mathbb{R}_+$ is any measurable cost function and $\Pi(P(X), P(Y))$ is the set of all the joint distributions $\pi(X, Y)$ whose marginals are $P(X)$ and $P(Y)$ respectively. The Wasserstein distance $W_c$ is then the "cost" of the optimal transport plan.

# 4 Measure evolving patterns via MI

## 4.1 EvoRate

Consider a sequence $\mathbf{z}_1^T : \{z_t\}_{t=1}^T$, a collection of sequential data points from time 1 to $T$, where each $z_t \in \mathbb{R}^D$ denotes a state or observation at the discrete time step $t$ with total $T$ steps. In practice, the sequence $\mathbf{z}_1^T$ can represent time series data, video, textual, audio, or any other ordered data stream.

We propose the use of the mutual information (MI) between the next observation and historical data over the past $k$ steps $I(\mathbf{Z}_{t-k+1}^t; Z_{t+1})$ to measure the evolving pattern within a time window of length $k$. In the literature, the mutual information is empirically estimated through equation 2, which involves learning the critic function $m$ [8, 11, 46, 36, 37]. However, one critical issue with existing works is that they ignore the temporal dependency of the data, and therefore the critic function $m$ can have a high bias for sequential data (shown in Figure 1a, 1b).

To take the temporal dependency into account when estimating $I(\mathbf{Z}_{t-k+1}^t; Z_{t+1})$, instead of learning the critic function $m$, we propose learning the autoregressive function $f$, which summarizes the historical information embedded in $Z_{t-k+1}^t$, and measuring its distance to $Z_{t+1}$ via the squared error metric. Specifically, we introduce **EvoRate** to estimate the empirical sequential MI $\hat{I}(\mathbf{Z}_{t-k+1}^t; Z_{t+1})$ by defining $m : \mathbb{R}^{k \times D} \times \mathbb{R}^D \to \mathbb{R}$, $m(x_1^k, y) = -\|f(g(x_1), .., g(x_k)) - g(y)\|_2^2$ in equation 2:

$$\text{EvoRate} := \hat{I}(\mathbf{Z}_{t-k+1}^t; Z_{t+1}) = \sup_{f,g} \mathbb{E}_{\mathbf{z}_{t-k+1}^{t+1} \sim \hat{P}(\mathbf{Z}_{t-k+1}, \ldots, Z_{t+1})} - \|f(g(z_{t-k+1}), \ldots, g(z_t)) - g(z_{t+1})\|_2^2$$

$$- \log \mathbb{E}_{\mathbf{z}_{t-k+1}^t \sim \hat{P}(Z_{t-k+1}, \ldots, Z_t), z_{t+1} \sim \hat{P}(Z_{t+1})} e^{-\|f(g(z_{t-k+1}), \ldots, g(z_t)) - g(z_{t+1})\|_2^2}, \qquad (4)$$

where $g : \mathbb{R}^D \to \mathbb{R}^d$ is an encoder. By selecting a different $d$, we can make a trade-off between computational cost and MI estimation accuracy. With $d \ll D$, EVORATE is a more computationally efficient method for approximating sequential MI than learning an autoregressive model in the original data space. However, due to the Data-processing inequality [12], this results in lower MI estimates. As $g$ is employed as an identity function, MI is estimated in the original space, thereby enhancing estimation correctness at the expense of increased computational consumption.

## 4.2 Discussion

In this section, we justify the validity of EVORATE as a metric of evolving patterns through the lens of a $k$-th order autoregression. Specifically, we define the Maximum likelihood estimation (MLE) loss as $\mathcal{L}_{mle} = -\mathbb{E}_{P(Z_{t+1}, \mathbf{z}_{t-k+1}^t)} \log Q(Z_{t+1}|\mathbf{Z}_{t-k+1}^t)$, where $Q$ is the probability distribution learned by the autoregressive model $F$ trained with a supervised loss (MLE, MSE) on sequential data. Note that the MLE loss can also be viewed as the expected risk of autoregressive prediction tasks [44].

The following proposition establishes the connection between the expected risk of a $k$-th order autoregression task and the mutual information $I(\mathbf{Z}_{t-k+1}^t; Z_{t+1})$:

**Proposition 1.** *Let $H$ denote the entropy. For autoregression tasks, the expected MLE loss satisfy:*

$$\mathcal{L}_{mle} = \underbrace{D_{\text{KL}}(P(Z_{t+1}|\mathbf{Z}_{t-k+1}^t), Q(Z_{t+1}|\mathbf{Z}_{t-k+1}^t))}_{(i) \ Model \ related} + \underbrace{H(Z_{t+1}) - I(Z_{t+1}; \mathbf{Z}_{t-k+1}^t)}_{(ii) \ Data \ related} \qquad (5)$$

A proof of the proposition is provided in Appendix A. Proposition 1 provides novel insights into learning a predictive model for an autoregression task from an information-theoretic perspective:

1. The expected risk can be decomposed into two orthogonal factors, where (i) measures the distance between the learned distribution $Q$ and true distribution $P$, and therefore is determined by the predictive model $F$. (ii) quantifies the inherent temporal dependency of the sequence. Notably, it is independent of $F$.

2. More importantly, when $Z$ is a discrete variable, due to the nature of mutual information, $I(Z_{t+1}; \mathbf{Z}_{t-k+1}^t) \leq H(Z_{t+1})$ and (i) attains a minimum of zero when the observed sequence $\mathbf{Z}_{t-k+1}^t$ encapsulates all the information of $Z_{t+1}$. Conversely, Proposition 1 reveals that even if $F$ can properly learn the true probability $P$ (i.e., (i) is small), its expected risk remains high when there is no temporal dependency that can be leveraged (i.e., $I$ is small).

Consequently, EvoRate, as an empirical estimate of $I(Z_{t+1}; \mathbf{Z}_{t-k+1}^t)$, can play an important role in indicating the success of learning from sequential data and therefore is adopted to quantify the evolving pattern in this work.

Subsequently, we demonstrate that MSE loss defined as $\mathcal{L}_{mse} = \mathbb{E}_{P(Z_{t+1}, \mathbf{z}_{t-k+1}^t)} || F(\mathbf{z}_{t-k+1}^t) - z_{t+1} ||_2^2$ can be interpreted as a variant of MLE loss, hence MI can be applied to a wide range of sequential data tasks that utilize MSE loss.

**Proposition 2.** *Assume that the predicted conditional probability density $Q$ learned by the prediction model follows $Q(Z_{t+1} | \mathbf{Z}_{t-k+1}^t) = \mathcal{N}(Z_{t+1} | F(\mathbf{Z}_{t-k+1}^t), I_D)$, where $\mathcal{N}(\cdot)$ denotes a Gaussian distribution with mean $F(\mathbf{Z}_{t-k+1}^t)$ and identity covariance matrix $I_D$. Then, the following holds*

$$\mathcal{L}_{mle} = \mathcal{L}_{mse} + const, \tag{6}$$

*where $\mathcal{L}_{mse}$ is the MSE loss and const is a constant term.*

## 5 Measure evolving patterns without correspondences

### 5.1 Estimate joint distribution

In many real-world applications, instead of processing many data point observations at different timestamps as data sequences, one needs to handle a data set at each timestamp: $\{z_{t,i}\}_{i=1}^{n_t}$ collected from multiple timestamps $t = \{1, \ldots, T\}$ [48, 42], where $i$ is the sample index and $n_t$ is the number of samples collected at timestamp $t$. The distribution $P(Z_t)$ associated with these data sets evolves over time $t \in \mathbb{R}$. For example, consider a supervised learning problem involving medical data $z_{t,i} = (x_{t,i}, y_{t,i})$ collected from multiple patients at different ages [42, 6]. In this scenario, we do not track the same patient across different ages, resulting in a lack of correspondence between timestamps and our objective extends to characterizing the evolving patterns of $\{Z_t\}_{t=1}^T$ across these discrete timestamps. However, EVORATE proposed in subsection 4.1 cannot be applied to this context due to the absence of the correspondences.

Estimating the mutual information from two data sets requires the pairwise correspondences between the sample of two data sets, which are assumed as given in existing works [8, 1, 27, 10, 29]. The correspondence between $Z_t$ and $Z_{t+1}$ reflects their joint distribution as it encapsulates how the values of $Z_t$ and $Z_{t+1}$ co-occur. This structured relationship indicates the interdependence of $Z_t$ and $Z_{t+1}$, which the joint distribution quantifies. Since the absence of the correspondence (i.e., an object observed at time $t$ is not at time $t + 1$), we can not access the joint probability distribution of the past states and the next state. To tackle this issue, we estimate the joint distribution through the optimal transport plan of the Wasserstein Distance. Specifically, we define the distance loss according to a joint distribution measurement $\pi$

$$\mathcal{L}_\mathcal{W}^t(\pi, f) = \mathbb{E}_{(z_t, z_{t+1}) \sim \pi} || f(g(z_t)) - g(z_{t+1}) ||_2^2 \tag{7}$$

where $g$ is fixed from updated gradients computed from $\mathcal{L}_\mathcal{W}^t$. Empirically, allowing $g$ to update during model training leads to the undesirable outcome of all representations collapsing into a single point as a result of minimizing the Wasserstein distance loss. To avoid this and preserve maximal information within the representations, $g$ is trained separately using an auto-encoder architecture with a reconstruction MSE loss.

Then, we compute the optimal transport plan $\pi^*$ to approximate the real joint distribution

$$\pi^*(Z_t, Z_{t+1}) = \underset{\pi \in \Pi(P(Z_t), P(Z_{t+1}))}{\arg\min} \mathcal{L}_\mathcal{W}^t(\pi, f), \quad \forall t \in \{1, \ldots, T-1\}, \tag{8}$$

and $f$ is updated in an alternating optimization manner with fixed $\pi^*$ to minimize $\mathcal{L}_\mathcal{W}^t(\pi^*, f)$. In practice, the following implementation is used: $\gamma^* := \arg\min_{\gamma \in \Pi(\hat{P}(Z_t), \hat{P}(Z_{t+1}))} \langle \mathcal{C}, \gamma \rangle_F$, where $\mathcal{C}$ is

the cost matrix with $\mathcal{C}_{i,j} = -\|g(z_{t,i}) - g(z_{t+1,j})\|_2^2$. When the original dimension $D$ is low, $g$ can be an identity function for precise MI estimation. Conversely, when $D$ is high, directly learning $f$ from $Z_t$ to $Z_{t+1}$ requires more accurate information and precise correspondence. This is because $f$ must be a considerably more complex and larger model to facilitate mapping from one high-dimensional space to another. As a result, the hypothesis space $\mathcal{F}$ for $f$ expands, requiring more information to ensure the model converges to an optimal state. The absence of correspondence therefore presents a challenge as it leads to an information-insufficient situation and it becomes more suitable to set a smaller representation dimension $d$.

It is noted that when the correspondences between two consecutive timestamps exist, they can be inferred by minimizing the Wasserstein distance. When such correspondences do not exist, one can still establish correspondences by identifying a proxy of $z_{t,i}$ in the succeeding timestamp that exhibits similar dynamics and shares the latent evolving patterns.

### 5.2 EVORATE$_\mathcal{W}$

We hence use $\pi^*(Z_t, Z_{t+1})$ to estimate joint distribution $P$, and then obtain the following estimator with $\pi^*(Z_t, Z_{t+1})$

$$\text{EvoRate}_\mathcal{W} = \sup_f \mathbb{E}_{(z_t, z_{t+1}) \sim \pi^*(Z_t, Z_{t+1})} - \|f(g(z_t))) - g(z_{t+1})\|_2^2$$
$$- \log \mathbb{E}_{z_t \sim \hat{P}(Z_t), z_{t+1} \sim \hat{P}(Z_{t+1})} e^{-\|f(g(z_t)) - g(z_{t+1})\|_2^2} \tag{9}$$

Here $k$ can be regarded to set to 1 compared to equation 4, indicating that EVORATE$_\mathcal{W}$ focuses on the first-order evolving patterns. It is possible to extend this approach to estimate higher-order $k$-order sequences by iteratively leveraging outcomes from first-order through to $k$-order sequential modeling.

### 5.3 Discussion

The following assumption argues that there exists an optimal function that precisely captures the underlying dynamics of evolving data.

**Assumption 1.** *(Realization) In machine learning prediction tasks, there exists a function $f^* \in \mathcal{F}$: $\mathcal{Z} \to \mathcal{Z}$ where the conditional distribution of $Z_{t+1}$ given $Z_t$ satisfies*

$$Z_{t+1} \sim \mathcal{N}(f^*(Z_t), \sigma^2 I) = P(Z_{t+1}|Z_t) \tag{10}$$

The following lemma demonstrates that when $f$ reaches the optimal predictive model $f^*$, the estimated optimal transport plan equals the real joint distribution. In this context, we consider $g$ to be the identity function.

**Lemma 1.** *Let $P(Z_t, Z_{t+1})$ be the ground truth joint distribution. If $f$ attains $f^*$, then*

$$\pi^*(Z_t, Z_{t+1}) = P(Z_t, Z_{t+1}) \tag{11}$$

Below, we give an illustrative example. As $T \gg 1$, the function $f$ will converge to $f^*$ by minimizing $\mathcal{L}_\mathcal{W}^t(\pi^*, f)$. It demonstrates that for a dynamic system without correspondences, the number of timestamps must be greater than 1 to learn the optimal autoregressive model effectively.

**Example** Consider data collected from multiple time steps where each sample is a vector $Z_t \in \mathbb{R}^D$. Specifically, the initial data points at the first timestamp is modeled as a Gaussian variable $Z_1 \sim \mathcal{N}(\mu_1, \Sigma_1)$. The temporal evolution of the data is governed by a transition function

$$z_{t+1} = f^*(Z_t) = A^* z_t + b^*, t \in \{1, \ldots, T\}$$

and each $Z_t$ follows a Gaussian distribution $Z_t \sim \mathcal{N}(\mu_t, \Sigma_t)$ where $A^* \in \mathbb{R}^{D \times D}, b^* \in R^D$. Solving the optimizing problem $\mathcal{L}_\mathcal{W}^t(\pi^*, f), t \in \{1, \ldots, T-1\}$ can lead to the solutions reaching optimal mapping $f^*$ with $t \gg 1$. (Experiment results shown in Figure 1c,1d)

## 6 Experiment

### 6.1 Multivariate Gaussians with tractable MI

**Sequential data with known correspondence** We sample data sequences $\{z_t\}_{t=1}^T, t \in \{1, \ldots, T\}$, $z_T = \rho \frac{\sum_{t=1}^{T-1} z_t}{T-1} + \sqrt{1-\rho^2}\epsilon$, with correlation $\rho \in [-1, 1]$, $\epsilon \sim \mathcal{N}(0, I)$, $Z_t \sim \mathcal{N}(0, I), t \in$

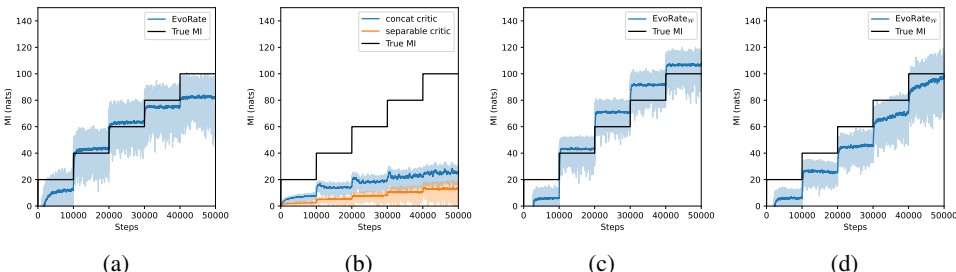

Figure 1: (a-b) Performance of (a) EVORATE / (b) concat and separate critic on mutual information estimation on sequential data with correspondence. (c-d) Performance of EVORATE$_\mathcal{W}$ on mutual information estimation on two consecutive time steps without correspondence, where $g$ is (c) an identity function / (d) neural nets.

$\{1, \ldots, T-1\}$. Given the correlation coefficient $\rho$ and dimensionality $D = 128$, we can compute the ground truth MI as $EvoRate(\mathbf{Z}_1^{T-1}; Z_T) = -(D/2)\ln(1-\rho^2)$. The optimal MI estimation can be achieved when sequential model $f$ equals the ground truth model $f^* = Avg$, where $Avg(\cdot)$ is an average operation. In Figure 1a and 1b, we increase $\rho$ over training steps to show the estimator behavior depends on the true mutual information. Additionally, we experiment with two forms of architecture: separable and joint. Separable architectures independently map the representations of history states $f(\mathbf{Z}_1^{T-1})$ and the future state $Z_T$ to an embedding space with neural nets $\phi_1$ and $\phi_2$ separably, and then take the inner product, i.e. $\phi_1(f(\mathbf{Z}_1^{T-1}))^T\phi_2(Z_T)$ as in [37]. Joint critics concatenate each $f(\mathbf{Z}_1^{T-1})$, $Z_T$ pair before feeding it into the network, i.e. $\phi([f(\mathbf{Z}_1^{T-1}); Z_T])$ as in [8]. In this experiment, $g$ is set to an identity function, and the sequential model $f$ is set to an LSTM [25]. All networks are fully-connected networks with ReLU activations. Figure 1a shows the estimated mutual information by EVORATE over the number of iterations, and square error metric can let $f$ converge to $f^*$ such that the EVORATE$_\mathcal{W}$ converges to ground truth mutual information. Figure 1b verifies that the square error metric has a lower bias compared to trainable concat critic and separable critic.

**Sequential data without known correspondence** We sample data sequence $\{z_t\}_{t=1}^T, t \in \{1, \ldots, T-1\}$, $z_{t+1} = \rho(A^*z_t + b^*) + \sqrt{1-\rho^2}\epsilon$, where $A^* \in \mathbb{R}^{D \times D}$ is a rotation matrix, $b^* \in \mathbb{R}^D$ is a translation vector, correlation of $\rho \in [-1, 1]$, $\epsilon \sim \mathcal{N}(0, I)$, and $Z_1 \sim \mathcal{N}(0, I)$. Given the correlation coefficient $\rho$ and dimensionality $D = 128$, we can compute the ground truth MI value $EvoRate(Z_t; Z_{t+1}) = -(D/2)\ln(1-\rho^2)$. The optimal MI estimation can be achieved when sequential model $f$ equals the ground truth model $f^* = A^*z_t + b^*$. In this experiment, it is actually very difficult to estimate mutual information without correspondence. As a result, the estimations by joint and separable critic do not converge and fail in the case without correspondence, which further shows the square error metric shows better performance than the trainable neural nets critic. In Figure 1c,1d, $g$ being an identity function estimates a higher value than $g$ being a neural-nets encoder. It is noted that EVORATE$_\mathcal{W}$ is the only method able to estimate the mutual information without the correspondence between timestamps, achieving a reasonable performance to estimate MI.

## 6.2 Sequential data's order approximation and feature selection

**Order Approximation** We sample data with 5-order ($k = 5$), and dimensionality $D = 5$, which means $Z_{t+1}$ is determined by $\mathbf{Z}_{t-4}^t$. More specifically, the data is generated by the dynamic function $Z_{t+1} = A^*\text{vec}(Z_{t-4}^t) + b^*$, where in this experiment, $\text{vec}(\cdot)$ is a vectorized operation, $A^* \in \mathbb{R}^{5 \times 25}$ and $b^* \in \mathbb{R}^5$. We vary $k \in \{1, 3, 5, 12, 24\}$ to measure the EVORATE between $\mathbf{Z}_{t-k+1}^t$ and $Z_{t+1}$. Figure 2a shows that $k = 5$ has the maximal EVORATE value. In another experiment, the time series forecasting task is used to verify the effectiveness of EVORATE. Time series forecasting performance is evaluated with the sMAPE metric [35], measured as the mean absolute error scaled by the magnitude of the predictions and target. The performance shown in Table 1 is the SOTA method [53] and they set the model with order $k = 45$. The order is set as $k \in \{10, 25, 45, 90, 180, 270\}$. Although $k = 270$ achieves the highest EvoRate, the difference between $k = 270$ and $k = 90$ is only 0.03, and the performances over average (AVG) sMAPE have the same prediction error. For M4-Weekly,

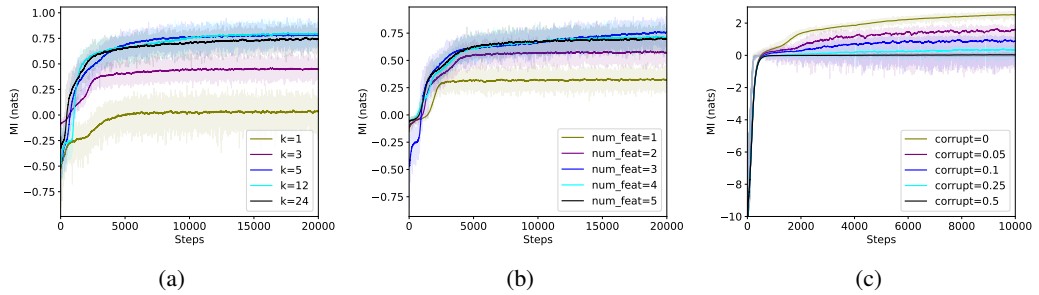

Figure 2: (a) $k$-order EVORATE estimation. (b) EVORATE estimation on a different number of features. (c) EVORATE estimation of the video prediction tasks with a different corruption rate.

EVORATE shows order set as $k = 90$ can achieve a good performance. Although EVORATE is slightly higher for $k = 270$ than $k = 90$, it sacrifices three times more computation consumptions compared to only a $+0.03$ EVORATE gain if the model time complexity is $\mathcal{O}(k)$. Forecastability (ForeCA) fails in this experiment, as shown in Table 1, since the longer time series shows smaller forecastability but it achieves smaller sMAPE and a better performance. Longer sequence can have more evolving patterns in different frequencies combined and result in a smaller forecastability, but it may be more easily predictable once the patterns are learned by the model. Therefore, we conclude that the entropy used by ForeCA is not a good indicator of the capability of predictions while MI used by EVORATE is. In addition, randomness is a critical factor for the capability of the predictions of the sequential data. Since one of the evolving patterns is learned by sequential models, the performance only relies on the randomness of the data, which can be regarded as unwanted noises or unobserved factors.

Table 1: **Time series forecasting (TSF) tasks: M4-Weekly** The values of EVORATE and time series forecasting performance below are experiments on dataset M4-Weekly. Here, short, medium, long, Avg stands for short-horizon sMAPE, medium-horizon sMAPE, long-horizon sMAPE, and the whole average sMAPE.

| ORDER:K | SHORT | MEDIUM | LONG | AVG | EVORATE | FORECA |
|---------|-------|--------|------|-------|---------|--------|
| 10      | 8.28  | 10.13  | 11.44| 10.06 | 1.98    | 0.50   |
| 25      | 5.78  | 9.82   | 10.85| 8.97  | 2.07    | 0.39   |
| 45      | 5.69  | 8.80   | 8.52 | 7.74  | 2.11    | 0.33   |
| 90      | 5.48  | 5.92   | 7.22 | 6.28  | 2.55    | 0.27   |
| 180     | 5.40  | 6.41   | 7.39 | 6.47  | 2.56    | 0.22   |
| 270     | 5.47  | 6.39   | 6.84 | 6.28  | 2.58    | 0.19   |

**Feature Selections** For autoregressive tasks, poor predictions may due to the lack of the features. Some features may be redundant and some may be unrelated to predictions. Others may be related to the task but are not put as input fed into the prediction model. The synthetic data has 5 dimensions, where the first 3 are useful, the fourth is redundant and the fifth is unrelated (Details in Appendix B.4). Figure 2b shows the EVORATE of the data sequence with the first $n$ features. The results show that i) EVORATE achieves the highest value with the first three features, ii) the first four features containing one redundant feature sees a minor performance drop, and iii) using all five features sees a larger drop.

### 6.3 EvoRate as a criterion for existence of evolving patterns

In some problems, data is sampled independently from the history observations [49, 2]. In this case, we suggest directly learning a model using ERM [49] for i.i.d (independent and identically distributed) or IRM [2] for data sampled independently but with distribution shifts. In many machine learning applications, data is predicted in an autoregressive manner by training sequential models [50, 9, 16, 33]. Whether to use ERM/IRM or sequential models directly depends on the existence of the evolving patterns. Therefore, we take EVORATE as the criterion for the existence of evolving patterns.

**Multivariate time series** In Table 2, EVORATE can achieve better estimates of the evolving patterns compared to ForeCA, where stronger evolving patterns indicate smaller regression errors using

sequential models. Specifically, for M4-Monthly and M4-daily, ForeCA shows equal values but EVORATE shows higher values for M4-Daily, consistent with experimental results in which M4-Daily achieves lower sMAPE.

Table 2: **Time series forecasting (TSF) tasks:** The estimated mutual information for the sequential data for different datasets. RMSE (Crypto, Player Traj.)/sMAPE (M4-Monthly, M4-Weekly, M4-Daily) is the performance of one SOTA TSF method [53].

|  | CRYPTO | PLAYER TRAJ. | M4- MONTHLY | M4- WEEKLY | M4- DAILY |
|---|---|---|---|---|---|
| RMSE/sMAPE | $6.91 \pm 0.01$ | $1.16 \pm 0.01$ | 11.93 | 7.25 | 2.99 |
| FORECA | 0.35 | 0.49 | 0.44 | 0.43 | 0.44 |
| EVORATE | 2.80 | 4.67 | 1.58 | 2.25 | 2.26 |

**Evolving Domain Generalization (EDG)** follows our setting in Section 5.1, where the correspondence is intractable and we aim to learn the evolving patterns to predict $y_{t,i}$ conditioned on input $x_{t,i}$ for every sample $z_{t,i} = \{x_{t,i}, y_{t,i}\}$ [42, 57]. In Table 3, we show the performance of EVORATE$_\mathcal{W}$ and SOTA performance for invariant learning [2] and evolving learning [42, 57, 59]. Although, performance of the evolving representation learning not only depends on the existence of the evolving patterns (shown by values of EVORATE$_\mathcal{W}$), EVORATE$_\mathcal{W}$ is still a critical factor in deciding whether to use sequential models. For example, PORTRAITS hahs the lowest EVORATE$_\mathcal{W}$ 0.25 and show the smallest improvement 3.7% of the performance of evolving learning than invariant learning, and RGaussian has the highest EVORATE$_\mathcal{W}$ 1.58 and show the largest improvement of the performance as 50.2%.

Table 3: The estimated mutual information for the evolving domains for different datasets. The reported results are the average accuracy of the multiple target domains.

|  | RGAUSSIAN | CIRCLE | SINE | RMNIST | PORTRAITS | CALTRAN | POWERSUPPLY |
|---|---|---|---|---|---|---|---|
| INVARIANT (ACC:%) | 47.5 | 51.3 | 63.2 | 39.0 | 85.4 | 64.1 | 70.8 |
| EVOLVING (ACC:%) | 97.7 | 73.8 | 71.4 | 46.4 | 89.1 | 70.6 | 75.7 |
| ACC$_{EVO}$ - ACC$_{INV}$ (%) | 50.2 | 22.5 | 8.2 | 7.4 | 3.7 | 6.5 | 4.9 |
| EVORATE$_\mathcal{W}$ | 1.58 | 0.58 | 0.54 | 0.95 | 0.25 | 0.28 | 0.46 |

## 6.4 Control randomness to corrupt evolving patterns

**Video prediction** aims to predict future video frames from the current ones. In this experiment, we evaluate EVORATE on the KITTI dataset [20], which contains 28 driving videos with a resolution of $375 \times 1242$. 24 videos in KITTI dataset are used for training. We verify the performance of EVORATE by shuffling the index of the sequential data with a certain corrupt probability, and this randomness will decrease the evolving patterns (Figure 3).

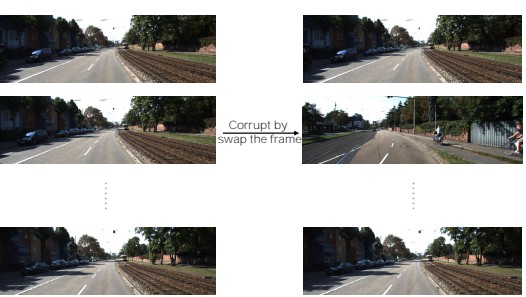

Figure 2c shows that by increasing the corruption rate to the video sequence, EVORATE exhibits a lower value. This is consistent with our

Figure 3: Illustration of corrupting the video's evolving patterns by randomly swapping the frame.

intuition, which is that the continuous video stream shows higher patterns compared to the disordered video clips.

## 6.5 Algorithms to improve performance on EDG tasks

From the intuition that $\mathcal{L}_\mathcal{W}$ can estimate the joint distribution, we apply this loss to learn a transition model based on the estimated joint distribution between two consecutive domains. Table 5 shows this is an efficient method and achieves improved performance on EDG tasks. Our method shows an 11.7% higher average accuracy than the second-best baselines.

Table 4: The comparison of the classification accuracy (%) between our and baseline methods across the synthetic and real-world datasets. The reported results are the average accuracy of the multiple target domains.

| ALGORITHM | MIXUP [55] | IRM [2] | CORAL [47] | DIVA [28] | LSSAE [43] | DRAIN [6] | OUR METHOD |
|---|---|---|---|---|---|---|---|
| RMNIST | 44.9 | 39.0 | 44.5 | 42.7 | 46.4 | 43.8 | **48.5** |
| RGAUSSIAN | 55.4 | 47.5 | 53.0 | 56.6 | 48.7 | 61.0 | **91.2** |
| POWERSUPPLY | 70.8 | 70.8 | 71.0 | 70.8 | 71.1 | 71.0 | **71.3** |
| AVG | 57.0 | 52.4 | 56.2 | 56.7 | 55.4 | 58.6 | **70.3** |

# 7  Conclusion

In this work, we propose EVORATE to qualitatively estimate the evolving patterns for the data sequences and the data snapshots from multiple consecutive timestamps without correspondences. We show the square error metric can be both a better critic for mutual information estimation, and a well-designed loss to help the optimal transport plan converge to the real joint distribution and the sequential model converge to the latent dynamic governing function. EVORATE reflects the complex patterns for high-dimensional data and is more computationally efficient than directly evaluating the performance of sequential data predictions. Experiments show EVORATE is an effective measure for evolving patterns and has the potential for many applications in the machine learning area.

## Acknowledgements

We appreciate constructive feedback from anonymous reviewers and meta-reviewers. This work is supported by the Natural Sciences and Engineering Research Council of Canada (NSERC), Discovery Grants program.

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

# A Proof of Theories

**Proposition 1.** *Assume that the model's probability density follows* $Q(Z_{t+1}|Z_{t-k+1}^t) = \mathcal{N}(Z_{t+1}; F(\mathbf{Z}_{t-k+1}^t), I_D)$

$$\mathcal{L}_{mle} = \mathcal{L}_{mse} + const \tag{12}$$

*Proof.*

$$\mathcal{L}_{mle} = -\mathbb{E}_P \log Q = ||F(\mathbf{z}_{t-k+1}^t) - z_{t+1}||_2^2 + \frac{D}{2}\log(2\pi) = \mathcal{L}_{mse} + const \tag{13}$$

$\square$

**Proposition 2.** *For autoregression tasks, the expected risk satisfy:*

$$\mathcal{L}_{mle} = -I(Z_{t+1}; \mathbf{Z}_{t-k+1}^t) + H(Z_{t+1}) + D_{\mathrm{KL}}(P(Z_{t+1}|\mathbf{Z}_{t-k+1}^t), Q(Z_{t+1}|\mathbf{Z}_{t-k+1}^t)) \tag{14}$$

*Proof.*

$$\mathcal{L}_{mle} = -\mathbb{E}_{\mathbf{z}_{t-k+1}^{t+1} \sim P(Z_{t-k+1},...,Z_{t+1})} \log Q(Z_{t+1}|\mathbf{Z}_{t-k+1}^t) \tag{15}$$

$$= -\mathbb{E}_{\mathbf{z}_{t-k+1}^{t+1} \sim P(Z_{t-k+1},...,Z_{t+1})} \log \frac{Q(Z_{t+1}|\mathbf{Z}_{t-k+1}^t)}{P(Z_{t+1}|\mathbf{Z}_{t-k+1}^t)}$$
$$- \mathbb{E}_{\mathbf{z}_{t-k+1}^{t+1} \sim P(Z_{t-k+1},...,Z_{t+1})} \log P(Z_{t+1}|\mathbf{Z}_{t-k+1}^t) \tag{16}$$

$$= D_{\mathrm{KL}}(P(Z_{t+1}|\mathbf{Z}_{t-k+1}^t), Q(Z_{t+1}|\mathbf{Z}_{t-k+1}^t)) - \mathbb{E}_{\mathbf{z}_{t-k+1}^{t+1} \sim P(Z_{t-k+1},...,Z_{t+1})} \log \frac{P(Z_{t+1}|\mathbf{Z}_{t-k+1}^t)}{P(Z_{t+1})}$$
$$- \mathbb{E}_{\mathbf{z}_{t-k+1}^{t+1} \sim P(Z_{t-k+1},...,Z_{t+1})} \log P(Z_{t+1}) \tag{17}$$

$$= D_{\mathrm{KL}}(P(Z_{t+1}|\mathbf{Z}_{t-k+1}^t), Q(Z_{t+1}|\mathbf{Z}_{t-k+1}^t)) - I(Z_{t+1}; \mathbf{Z}_{t-k+1}^t) + H(Z_{t+1}) \tag{18}$$

$\square$

$\beta$-mixing is a measure of the degree of dependence between random variables in a sequence over time, which is closely related to MI and furthermore upper bounded by MI:

**Remark 1.** $s \in \mathbb{N}$, $\beta$-*mixing coefficients defined in below satisfy:*

$$\beta(s) = \sup_s \mathbb{E}_{\mathbf{Z}_{-\infty}^t} \left[ ||P_{t+s}^\infty(\cdot|\mathbf{Z}_{-\infty}^t) - P_{t+s}^\infty||_{TV} \right] \le \sup_s \left[ \sqrt{2I(Z_{t+s}^\infty; \mathbf{Z}_{-\infty}^t)} \right] \tag{19}$$

*where* $|| \cdot ||_{TV}$ *is the maximum total variation distance.*

*Proof.*

$$\beta(s) = \sup_s \mathbb{E}_{\mathbf{Z}_{-\infty}^t} \left[ ||P_{t+s}^\infty(\cdot|\mathbf{Z}_{-\infty}^t) - P_{t+s}^\infty||_{TV} \right]$$

$$\le \sup_s \mathbb{E}_{\mathbf{Z}_{-\infty}^t} \left[ \sqrt{2D_{\mathrm{KL}}(P(Z_{t+s}^\infty|\mathbf{Z}_{-\infty}^t)||P(Z_{t+s}^\infty))} \right] \tag{20}$$

$$\le \sup_s \left[ \sqrt{2\mathbb{E}_{\mathbf{Z}_{-\infty}^t} D_{\mathrm{KL}}(P(Z_{t+s}^\infty|\mathbf{Z}_{-\infty}^t)||P(Z_{t+s}^\infty))} \right] = \sup_s \left[ \sqrt{2I(Z_{t+s}^\infty; \mathbf{Z}_{-\infty}^t)} \right] \tag{21}$$

where the first inequality follows Pinsker's inequality; the second inequality follows Jensen's Inequality. $\square$

**Lemma 1.** $P(X, Y)$ *is the real underlying distribution, and* $\pi(X, Y)$ *is the optimal transport plan that satisfies both margins comply with* $P(X)$ *and* $P(Y)$. *We first define:*

$$I_P(X; Y) = \int p(x, y) \log \frac{p(x, y)}{p(x)p(y)} dx \, dy \tag{22}$$

$$I_\pi(X; Y) = \int \pi(x, y) \log \frac{\pi(x, y)}{p(x)p(y)} dx \, dy \tag{23}$$

*Then, we have*

$$I_p(X;Y) - I_\pi(X;Y) = H_\pi(X,Y) - H_p(X,Y) = H_\pi(Y|X) - H_p(Y|X) \tag{24}$$

*Therefore, we can get $I_P(Z_t; Z_{t+1}) - I_\pi(Z_t; Z_{t+1}) = H_\pi(Z_t; Z_{t+1}) - H_P(Z_t; Z_{t+1})$.*

*Proof.*

$$I_p(X;Y) - I_\pi(X;Y) = \int p(x,y) \log \frac{p(x,y)}{p(x)p(y)} dx\, dy - \int \pi(x,y) \log \frac{\pi(x,y)}{p(x)p(y)} dx\, dy \tag{25}$$

$$= \int p(x,y) \log p(x,y) dx\, dy - \int \pi(x,y) \log \pi(x,y) dx\, dy - \int p(x,y) \log p(x) dx\, dy$$

$$- \int p(x,y) \log p(y) dx\, dy + \int \pi(x,y) \log p(x) dx\, dy + \int \pi(x,y) \log p(y) dx\, dy \tag{26}$$

$$= \int p(x,y) \log p(x,y) dx\, dy - \int \pi(x,y) \log \pi(x,y) dx\, dy - \int p(x) \log p(x) dx\, dy$$

$$- \int p(y) \log p(y) dx\, dy + \int p(x) \log p(x) dx\, dy + \int p(y) \log p(y) dx\, dy \tag{27}$$

$$= H_\pi(X,Y) - H_p(X,Y) \tag{28}$$

$$= H_\pi(X|Y) - H_p(X|Y) = H_\pi(Y|X) - H_p(Y|X) \tag{29}$$

26 to 27 is due to $p(x,y)$ and $\pi(x,y)$ have the same margins $p(x)$ and $p(y)$ and we integral over $x$ or $y$ first. $\qquad\square$

**Lemma 2.** *Let $P(Z_t, Z_{t+1})$ be the ground truth joint distribution. If $f$ attains $f^*$, then*

$$\pi^*(Z_t, Z_{t+1}) = P(Z_t, Z_{t+1}) \tag{30}$$

*Proof.* To finish the proof, we first assume $(z_t, z'_{t+1}) \sim \pi(Z_t, Z_{t+1})$, $(z_t, z_{t+1}) \sim \pi(Z_t, Z_{t+1})$ and $\epsilon \sim \mathcal{N}(0, \sigma^2 I)$, then,

$$\inf_{\pi \in \Pi(P(Z_t), P(Z_{t+1}))} \mathcal{L}^t_\mathcal{W}(\pi, f) = \inf_{\pi \in \Pi(P(Z_t), P(Z_{t+1}))} \mathbb{E}_{(z_t, z'_{t+1}) \sim \pi} ||f(z_t) - z'_{t+1})||_2^2$$

$$= \inf_{\pi \in \Pi(P(Z_t), P(Z_{t+1}))} \mathbb{E}_{(z_t, z'_{t+1}) \sim \pi} ||(z_{t+1} - \epsilon) - z'_{t+1})||_2^2$$

$$= \inf_{\pi \in \Pi(P(Z_t), P(Z_{t+1}))} \left[ \int ||z'_{t+1} - z_{t+1}||_2^2\, d\pi(Z_t, Z_{t+1}) \right. \tag{31}$$

$$\left. - \int 2\epsilon^T (z'_{t+1} - z_{t+1})\, d\pi(Z_t, Z_{t+1}) + \int ||\epsilon||_2^2\, d\pi(Z_t, Z_{t+1}) \right]$$

$$= \inf_{\pi \in \Pi(P(Z_t), P(Z_{t+1}))} \left[ \int ||z'_{t+1} - z_{t+1}||_2^2\, d\pi(Z_t, Z_{t+1}) \right. \tag{32}$$

$$- \int 2\epsilon^T (z'_{t+1} - f^*(z_t) + \epsilon)\, d\pi(Z_t, Z_{t+1}) \tag{33}$$

$$\left. + \int ||\epsilon||_2^2\, d\pi(Z_t, Z_{t+1}) \right]$$

$$= \inf_{\pi \in \Pi(P(Z_t), P(Z_{t+1}))} \left[ \int ||z'_{t+1} - z_{t+1}||_2^2\, d\pi(Z_t, Z_{t+1}) \right. \tag{34}$$

$$\left. - \int 2\epsilon^T (z'_{t+1} - f^*(z_t))\, d\pi(Z_t, Z_{t+1}) - \int ||\epsilon||_2^2\, d\pi(Z_t, Z_{t+1}) \right].$$

Since $\epsilon \perp\!\!\!\perp (z'_{t+1} - f^*(z_t))$

$$\inf_{\pi \in \Pi(P(Z_t), P(Z_{t+1}))} \mathcal{L}^t_\mathcal{W}(\pi, f) = \inf_{\pi \in \Pi(P(Z_t), P(Z_{t+1}))} \int ||z'_{t+1} - z_{t+1}||_2^2\, d\pi(Z_t, Z_{t+1})$$

$$+ const$$

To achieve infimum of $\mathcal{L}^t_\mathcal{W}(\pi, f)$, $Z'_{t+1} = Z_{t+1}$ should satisfy and hence $\pi^*(Z_{t+1}|Z_t) = P(Z_{t+1}|Z_t)$ and $\pi^*(Z_{t+1}, Z_t) = P(Z_{t+1}, Z_t)$ with a feasible solution. $\qquad\square$

**Example** We consider data collected from multiple time steps where each sample is a vector $Z_t \in \mathbb{R}^D$. Specifically, the initial data points at the first timestamp is modeled as a Gaussian variable $Z_1 \sim \mathcal{N}(\mu_1, \Sigma_1)$. The temporal evolution of the data is governed by a transition function:

$$z_{t+1} = f^*(Z_t) = A^* z_t + b^*, t \in \{1, \ldots, T\},$$

and each $Z_t$ follows a Gaussian distribution, $Z_t \sim \mathcal{N}(\mu_t, \Sigma_t)$, where $A^* \in \mathbb{R}^{D \times D}$, $b^* \in R^D$. Solving the optimizing problem $\mathcal{L}_{\mathcal{W}}^t(\pi, f), t \in \{1, \ldots, T-1\}$ can lead to the solutions reaching optimal mapping $f^*$ with $t \gg 1$.

*Proof.* Assume the linear transition function has parameters $(A, b)$ as $z_{t+1} = f(Z_t) = Az_t + b$, it can be inferred that $Z_{t+1} \sim \mathcal{N}(A\mu_t + b, A\Sigma_t A^T)$.

For each pair of data from two consecutive timestamp data, the Wasserstein distance loss can be expressed as follows, according to [21]

$$\text{Wasserstein Distance loss} = \inf_\pi \mathcal{L}_{\mathcal{W}}(t) = ||\mu_t - (A\mu_{t-1} + b)||_2^2 +$$

$$Tr\left(A\Sigma_{t-1}A^T + \Sigma_t - 2\big(\Sigma_t^{\frac{1}{2}} A\Sigma_{t-1}A^T \Sigma_t^{\frac{1}{2}}\big)^{\frac{1}{2}}\right)$$

Then, we can have

$$\inf_\pi \mathcal{L}_{\mathcal{W}}(t) = ||\mu_t - (A\mu_{t-1} + b)||_2^2 + Tr\left(A\Sigma_{t-1}A^T + \Sigma_t\right) - \left(2Tr\big(\Sigma_t^{\frac{1}{2}} A\Sigma_{t-1}A^T \Sigma_t^{\frac{1}{2}}\big)^{\frac{1}{2}}\right)$$

$$= ||\mu_t - (A\mu_{t-1} + b)||_2^2 + Tr\left(A\Sigma_{t-1}A^T + \Sigma_t\right) - \left(2Tr\big(A^T \Sigma_t A\Sigma_{t-1}\big)^{\frac{1}{2}}\right)$$

$$= ||\mu_t - (A\mu_{t-1} + b)||_2^2 + Tr\left(A\Sigma_{t-1}A^T + \Sigma_t - 2\big(A^T \Sigma_t A\Sigma_{t-1}\big)^{\frac{1}{2}}\right)$$

$$= ||\mu_t - (A\mu_{t-1} + b)||_2^2 + Tr\left(A\Sigma_{t-1}A^T + \Sigma_t\right) - \left(2Tr\big(A^T \Sigma_t A\Sigma_{t-1}\big)^{\frac{1}{2}}\right)$$

$$= ||\mu_t - (A\mu_{t-1} + b)||_2^2 + Tr\left(A\Sigma_{t-1}A^T + \Sigma_t - 2\big(A^T \Sigma_t A\Sigma_{t-1}\big)^{\frac{1}{2}}\right)$$

$$= ||\mu_t - (A\mu_{t-1} + b)||_2^2 + Tr\left((A^T \Sigma_t A - \Sigma_{t-1})(A^T \Sigma_t A - \Sigma_{t-1})^T\right)$$

$$= ||\mu_t - (A\mu_{t-1} + b)||_2^2 + ||A^T \Sigma_t A - \Sigma_{t-1}||_F$$

where $|| \cdot ||_F$ is Frobenius norm. $\inf_\pi \mathcal{L}_{\mathcal{W}}(t) = 0$, and the infimum is attained when

$$\begin{cases} A\mu_{t-1} + b = \mu_t \\ A^T \Sigma_{t-1} A = \Sigma_t \end{cases}, \forall \, t = \{2, \ldots, T\} \tag{35}$$

We are dealing with a system where the matrix $A$ and $b$ together comprise $n \times (n+1)$ unknown variables. At each time step $t$, the system is described by $n \times n$ quadratic equations and $n$ linear equations. Typically, these quadratic equations yield two possible sets of solutions. To refine our estimates and converge towards the optimal parameters $(A^*, b^*)$, employing a large number of time steps ($t \gg 1$) allows us to formulate an overdetermined system of equations.

$\square$

# B  Datasets

## B.1  Multivariate Gaussians: sequential data with known correspondence

We sample data sequence $\{z_t\}_{t=1}^T, t \in \{1, \ldots, T\}$, $z_T = \rho \frac{\sum_{t=1}^{T-1} z_t}{T-1} + \sqrt{1 - \rho^2}\epsilon$, where correlation of $\rho \in [-1, 1]$, $\epsilon \sim \mathcal{N}(0, I)$, $Z_t \sim \mathcal{N}(0, I), t \in \{1, \ldots, T-1\}$. Given the correlation coefficient $\rho$ and dimensionality $D = 128$, we can compute the ground truth MI value $EvoRate(\mathbf{Z}_1^{T-1}; Z_T) = -(D/2)\ln(1 - \rho^2)$.

## B.2 Multivariate Gaussians: sequential data without known correspondence

We sample data sequence $\{z_t\}_{t=1}^T, t \in \{1, \ldots, T-1\}$, $z_{t+1} = \rho(A^* z_t + b^*) + \sqrt{1 - \rho^2}\epsilon$, where $A^* \in \mathbb{R}^{D \times D}$ is a rotation matrix, $b^* \in \mathbb{R}^D$ is a translation vector, correlation of $\rho \in [-1, 1]$, $\epsilon \sim \mathcal{N}(0, I)$, and $Z_1 \sim \mathcal{N}(0, I)$. Given the correlation coefficient $\rho$ and dimensionality $D = 128$, we can compute the ground truth MI value $EvoRate(Z_t; Z_{t+1}) = -(D/2)\ln(1 - \rho^2)$.

## B.3 Synthetic data for order approximation

We sample data with 5-order ($k = 5$), and dimensionality $D = 5$, which means $Z_{t+1}$ is determined by $Z_{t-4}^t$. More specifically, the data is generated by the dynamic function $Z_{t+1} = A^* \text{vec}(Z_{t-4}^t) + b^*$, where in this experiment, $\text{vec}(\cdot)$ is a vectorized operation, $A^* \in \mathbb{R}^{5 \times 25}$, and $b^* \in \mathbb{R}^5$. In this experiment, we set $k$ to $\{1, 3, 5, 12, 24\}$ to measure the EVORATE between $\mathbf{Z}_{t-k+1}^t$ and $Z_{t+1}$.

## B.4 Synthetic data for feature selection

In this experiment, We sample data with 5-order ($k = 5$), and dimensionality $D = 5$. Only first three features are decided by the past states: $Z_{t+1}[: 3] = A^* \text{vec}(Z_{t-4}^t[:, : 3]) + b^*$, where in this experiment, $\text{vec}(\cdot)$ is a vectorized operation, $A^* \in \mathbb{R}^{3 \times 15}$, and $b^* \in \mathbb{R}^3$. The fourth feature is a linear combination of the first three dimension features, as a redundant feature. The fifth feature is purely noise following the normal distribution.

## B.5 Time-series forecastings

**M4** [35] contains 10000 highly nonstationary univariate time series with different frequencies from hourly to yearly and different categories from financials to demographics. The forecasting horizon varies across different frequencies.

**Crypto** [4] This multivariate dataset contains 8 features on historical trades, such as open and close prices, for 14 cryptocurrencies. The original challenge is to predict 3-step ahead 15-minute relative future returns. Since we focus on long-term forecasting, we train all models to make 15-step predictions of 15-minute relative future returns. We use the original training set from the competition and do an 80%-10%-10% training-validation-test split.

**Player Trajectory** [32] contains basketball player movement trajectories from NBA games in 2016. We randomly sample 300 player trajectories for training and validation and 50 trajectories for testing. All models are trained to yield 30-step ahead predictions

## B.6 Evolving domain generalization

**Rotated Gaussian** [58] consists of 30 domains generated by the same Gaussian distribution, but the decision boundary rotates from $0°$ to $338°$ with an interval of $12°$. We split the domains into source domains (1-22 domains), intermediate domains (22-25 domains), and target domains (26-30 domains). The intermediate domains are utilized as the validation set.

**Circle** [40] contains evolving 30 domains where the instance are sampled from 30 2D Gaussian distributions. The label is assigned using a half-circle curve as the decision boundary. (15 source domains, 5 validation domains, and 10 target domains)

**Sine** In Sine [40] each data owns two attributes $(x_1, x_2)$. The label is assigned using a sine curve as the decision boundary. We rearrange this dataset by extending it to 24 evolving domains. Each domain covers $\frac{1}{24}$ the period of the sinusoid. (12 source domains, 4 validation domains, and 8 target domains)

**Rotated MNIST (RMNIST)** [22] is an adaptation of the popular MNIST digit dataset [15], composed of MNIST digits of various rotations. The task is to classify a digit from 0 to 9 given an image of the digit. We follow [43] and extend it to 19 evolving domains via applying the rotations with degree of $\{0°, 15°, 30°, \ldots, 180°\}$ in order. (10 source domains, 3 validation domains, and 6 target domains).

**Portraits** [23] is a real-world dataset that comprises photographs of American high school seniors collected over a period of 108 years (1905-2013) across 26 states. The objective is to accurately

classify the gender for each photograph. The dataset is divided into 34 domains based on a fixed interval over time. (19 source domains, 5 validation domains, and 10 target domains)

**Caltran** [26] consists of real-world images captured by a fixed traffic camera deployed in an intersection over time. Frames were updated at 3-minute intervals each with a resolution $320 \times 320$. We divide it into 34 domains by time. The task of Caltran is to classify scenes to identify the presence of one or more vehicles in or approaching the intersection. The challenge mainly raise from the continually evolving domain shift as changes include time, illumination, weather, etc. (19 source domains, 5 validation domains, and 10 target domains)

**PowerSupply** [14] is a dataset designed for the task of time-section prediction of current power supply based on hourly records obtained from an Italian electricity company. The dataset consists of 30 domains formed according to days. Each data point is assigned a binary class label indicating whether the current power supply belongs to the morning or the afternoon. Domain shifts may arise due to variations in season, weather, price, or the differences between working days and weekends. (15 source domains, 5 validation domains, and 10 target domains)

## C  Full experiments on EDG tasks

In this section, we present complete experimental results to validate the efficacy of our proposed evolving domain generalization task.

Table 5: RMNIST. We show the results on each target domain by domain index.

| Algorithm | 130° | 140° | 150° | 160° | 170° | 180° | Avg |
|---|---|---|---|---|---|---|---|
| Mixup [55] | $61.3 \pm 0.7$ | $47.4 \pm 0.8$ | $39.1 \pm 0.7$ | $38.3 \pm 0.7$ | $40.5 \pm 0.8$ | $42.8 \pm 0.9$ | 44.9 |
| IRM [2] | $47.7 \pm 0.9$ | $38.5 \pm 0.7$ | $34.1 \pm 0.7$ | $35.7 \pm 0.8$ | $37.8 \pm 0.8$ | $40.3 \pm 0.8$ | 39.0 |
| CORAL [47] | $58.8 \pm 0.9$ | $46.2 \pm 0.8$ | $38.9 \pm 0.7$ | $38.5 \pm 0.8$ | $41.3 \pm 0.8$ | $43.5 \pm 0.8$ | 44.5 |
| DIVA [28] | $58.3 \pm 0.8$ | $45.0 \pm 0.8$ | $37.6 \pm 0.8$ | $36.9 \pm 0.7$ | $38.1 \pm 0.8$ | $40.1 \pm 0.8$ | 42.7 |
| LSSAE [43] | $64.1 \pm 0.8$ | $51.6 \pm 0.8$ | $43.4 \pm 0.8$ | $38.6 \pm 0.7$ | $40.3 \pm 0.8$ | $40.4 \pm 0.8$ | 46.4 |
| DRAIN [6] | $59.5 \pm 0.8$ | $45.4 \pm 0.8$ | $40.2 \pm 0.7$ | $37.2 \pm 0.7$ | $39.6 \pm 0.8$ | $41.0 \pm 0.7$ | 43.8 |
| Our method | $65.5 \pm 0.6$ | $55.9 \pm 0.8$ | $47.3 \pm 0.8$ | $41.8 \pm 0.9$ | $40.1 \pm 0.9$ | $40.3 \pm 0.8$ | 48.5 |

Table 6: Rotated Gaussian. We show the results on each target domain by domain index.

| Algorithm | 26 | 27 | 28 | 29 | 30 | Avg |
|---|---|---|---|---|---|---|
| Mixup | $56.2 \pm 1.5$ | $63.4 \pm 3.0$ | $56.8 \pm 1.4$ | $49.4 \pm 1.5$ | $41.4 \pm 2.0$ | 55.4 |
| IRM | $56.8 \pm 1.9$ | $55.8 \pm 3.1$ | $51.8 \pm 2.3$ | $41.6 \pm 1.6$ | $31.4 \pm 2.1$ | 47.5 |
| CORAL | $54.8 \pm 1.6$ | $54.0 \pm 0.6$ | $53.8 \pm 1.0$ | $52.0 \pm 0.8$ | $50.6 \pm 1.6$ | 53.0 |
| DIVA | $59.0 \pm 1.5$ | $55.8 \pm 0.9$ | $53.6 \pm 0.7$ | $59.2 \pm 1.3$ | $55.6 \pm 1.5$ | 56.6 |
| LSSAE | $50.6 \pm 0.9$ | $50.8 \pm 2.3$ | $43.4 \pm 1.4$ | $48.4 \pm 2.4$ | $50.4 \pm 2.1$ | 48.7 |
| DRAIN | $73.2 \pm 2.9$ | $70.0 \pm 1.7$ | $63.8 \pm 2.4$ | $53.2 \pm 2.2$ | $45.0 \pm 1.2$ | 61.0 |
| Our method | $98.0 \pm 0.6$ | $94.6 \pm 0.9$ | $98.0 \pm 0.6$ | $92.4 \pm 0.9$ | $73.2 \pm 0.8$ | 91.2 |

Table 7: PowerSupply. We show the results on each target domain by domain index.

| Algorithm | 21 | 22 | 23 | 24 | 25 | 26 | 27 | 28 | 29 | 30 | Avg |
|---|---|---|---|---|---|---|---|---|---|---|---|
| Mixup | $69.6 \pm 1.4$ | $69.5 \pm 1.5$ | $68.3 \pm 1.5$ | $64.3 \pm 1.5$ | $87.1 \pm 1.0$ | $76.6 \pm 1.3$ | $70.1 \pm 1.4$ | $69.2 \pm 1.3$ | $68.1 \pm 1.5$ | $65.0 \pm 1.6$ | 70.8 |
| IRM | $69.8 \pm 1.4$ | $69.5 \pm 1.4$ | $68.3 \pm 1.4$ | $64.1 \pm 1.4$ | $87.2 \pm 0.9$ | $76.5 \pm 1.3$ | $70.0 \pm 1.5$ | $69.1 \pm 1.5$ | $68.2 \pm 1.3$ | $65.0 \pm 1.4$ | 70.8 |
| CORAL | $69.9 \pm 1.4$ | $69.7 \pm 1.4$ | $68.9 \pm 1.4$ | $64.6 \pm 1.4$ | $86.1 \pm 1.0$ | $76.3 \pm 1.3$ | $70.0 \pm 1.5$ | $69.5 \pm 1.5$ | $68.8 \pm 1.3$ | $65.7 \pm 1.5$ | 71.0 |
| DIVA | $69.7 \pm 1.4$ | $69.5 \pm 1.3$ | $68.2 \pm 1.4$ | $63.9 \pm 1.5$ | $87.5 \pm 1.0$ | $76.5 \pm 1.3$ | $69.9 \pm 1.5$ | $69.1 \pm 1.5$ | $68.1 \pm 1.3$ | $64.7 \pm 1.5$ | 70.7 |
| LSSAE | $70.0 \pm 1.4$ | $69.8 \pm 1.4$ | $69.0 \pm 1.5$ | $65.4 \pm 1.4$ | $85.1 \pm 1.1$ | $76.0 \pm 1.4$ | $70.1 \pm 1.7$ | $69.9 \pm 1.3$ | $69.0 \pm 1.6$ | $66.3 \pm 1.4$ | 71.1 |
| DRAIN | $70.1 \pm 1.3$ | $70.0 \pm 1.0$ | $69.3 \pm 1.1$ | $65.5 \pm 1.5$ | $83.6 \pm 1.0$ | $75.8 \pm 1.7$ | $70.3 \pm 1.3$ | $69.8 \pm 1.5$ | $68.9 \pm 1.9$ | $66.4 \pm 1.2$ | 71.0 |
| Our method | $69.4 \pm 1.3$ | $69.3 \pm 1.7$ | $68.2 \pm 1.3$ | $64.1 \pm 1.5$ | $86.2 \pm 1.0$ | $75.8 \pm 1.4$ | $70.3 \pm 1.2$ | $70.8 \pm 1.4$ | $70.0 \pm 1.5$ | $68.6 \pm 1.0$ | 71.3 |

## D Computational Cost

All experiments are carried out on 498G memory, 2 x AMD Milan 7413 @ 2.65 GHz 128M cache L3, and 2 x NVidia A100SXM4 (40 GB memory). The algorithm's computational cost is the cost of OT (using package [19]) $\mathcal{O}(B^3)$ and the cost of estimation of MI $\mathcal{O}(B^2d)$, where $B$ is the batch size in an iteration, and $d$ is the representation dimension. In all, the total computational cost is $\mathcal{O}(B^2d + B^3)$ (not counting the encoder $g$ and decoder $h$).

## E Algorithms Training Procedures

---
**Algorithm 1** EVORATE: Data is sampled in a sequential manner with correspondence

---
1: **for** each training iteration **do**
2:      Sample $\left\{ \{z_{t,i}\}_{t=1}^{T} \right\}_{i=1}^{B}$ from $p(\mathbf{z}_{t=1}^{T})$, where $B$ is the batch size per training iteration
3:      Compute EvoRate of $i$-th sample at timestamp $t$ according to Eq (4), $k < t$:

$$\text{EvoRate}_{t,i} := - ||f(g(z_{t-k+1,i}), \ldots, g(z_{t,i})) - g(z_{t+1,i})||_2^2$$

$$- \log \frac{1}{B} \sum_{j=1, j\neq i}^{B} e^{-||f(g(z_{t-k+1,i}), \ldots, g(z_{t,i})) - g(z_{t+1,j})||_2^2}$$

4:      Update $f$ and $g$ by maximize $\frac{1}{B(T-k)} \sum_{t=k}^{T-1} \sum_{i=1}^{B} \text{EvoRate}_{t,i}$
5: **end for**

---

---
**Algorithm 2** EVORATE$_{\mathcal{W}}$: Data is sampled from different timestamps but without correspondence

---
1: **for** each training iteration **do**
2:      Sample $\left\{ \{z_{t,i}\}_{i=1}^{B} \right\}_{t=1}^{T}$ from $p(\mathbf{z}_{t=1}^{T})$, where $B$ is the batch size per training iteration
3:      Compute the optimal transport plan $\pi^*$, where $\mathcal{L}_{\mathcal{W}}^t(\pi, f)$ defined in Eq (7)

$$\pi^*(Z_t, Z_{t+1}) = \mathop{\arg\min}_{\pi \in \Pi(P(Z_t), P(Z_{t+1}))} \mathcal{L}_{\mathcal{W}}^t(\pi, f), \quad \forall t \in \{1, \ldots, T-1\},$$

4:      Compute EvoRate$_{\mathcal{W}}$ of $i$-th sample at timestamp $t$ according to Eq (9), especially $(z_{t,i}, z_{t+1,i})$ is sampled from $\pi^*$:

$$(\text{EvoRate}_{\mathcal{W}})_{t,i} := -||f(g(z_{t,i})) - g(z_{t+1,i})||_2^2 - \log \frac{1}{B} \sum_{j=1, j\neq i}^{B} e^{-||f(g(z_{t,i})) - g(z_{t+1,j})||_2^2}$$

5:      Update $f$ by maximize $\frac{1}{B(T-1)} \sum_{t=1}^{T-1} \sum_{i=1}^{B} (\text{EvoRate}_{\mathcal{W}})_{t,i}$
6: **end for**

---

## F Comparisons with Traditional time-series statistic indicators

There are traditional statistic indicators for time series, but they have significant limitations: a) System sensitivity, measured by Lyapunov Exponents (LEs) [2] measures how sensitive a dynamic system is to initial conditions, b) Trend is the slope of a linear regression fitted to sequential data, and c) Seasonality, measured by the ACF test [3], assesses linear correlations between observations at different time lags. These methods are not designed to measure evolving patterns and struggle to handle high-dimensional data. Each method measures only one aspect of evolving patterns: system sensitivity, trend, or seasonality. Together, they determine the overall evolving patterns. We present a comparison of these metrics with our method below:

In the above table, a larger EvoRate consistently indicates a smaller potential prediction error (RMSE/sMAPE) for the dataset. In contrast, LEs, trend, and seasonality show little impact on the prediction errors. Another significant drawback of these methods is their inability to be directly applied to high-dimensional data, such as images, videos, and NLP datasets.

Table 8: Comparison of Different Metrics Across Various Datasets

|  | Crypto | Player Traj. | M4-Monthly | M4-Weekly | M4-Daily |
|---|---|---|---|---|---|
| RMSE/sMAPE | 6.91 | 1.16 | 11.93 | 7.25 | 2.99 |
| LEs | 0.026 | 0.052 | 0.011 | 0.013 | 0.020 |
| Trend | 0.02 | 0.01 | 0.48 | 0.13 | 0.05 |
| Seasonality | 0.00% | 0.00% | 66.34% | 0.00% | 0.00% |
| EvoRate | 2.80 | 4.67 | 1.58 | 2.25 | 2.26 |

## G   Limitations

Due to computational resource limitations, we have not included experiments involving Natural Language Processing (NLP) tasks on Large Language Models (LLMs) in our study. These models typically require extensive processing power and substantial memory, which exceed our current hardware capabilities. Additionally, the high costs associated with running these models make them impractical for our budget. Instead, we focused on alternative datasets and models that align with our available resources. We believe that our chosen datasets still provide valuable insights while remaining within our operational constraints. Future work could explore LLMs as our computational capacity expands.

