# OpenReview forum: "Towards Understanding Evolving Patterns in Sequential Data"
_NeurIPS.cc/2024/Conference — NeurIPS 2024 spotlight_

### Official Review · Reviewer_V8vA · 2024-06-21

**Soundness:** 3
**Presentation:** 3
**Contribution:** 3
**Rating:** 6
**Confidence:** 2

**Summary:**

This paper proposes a novel metric, Evolving Rate, using mutual information to measure the existence of the evolving patterns in sequential data. For the scenarios in which data samples across disparate time steps are not aligned, the paper proposes to build the correspondence between snapshots using optimal transport, and thus develop the corresponding EvoRate_w, to measure evolving patterns without correspondences. The final experiments conducted on multiple tasks using several datasets, demonstrate the superior performance of the proposed EvoRate and EvoRate using OT to align data across time steps, compared to SOTA baselines.

**Strengths:**

Significance:
Evolving patterns are important for us to use sequential models to observed time series data. This paper develop the evolving rate (EvoRate), to quantify the evolving patterns. In particular, this EvoRate can be to assess temporal order and to conduct feature selections in sequential observations.

Clarity:
The paper is well motivated! Most of the technical parts of the work are clearly presented!

Quality:
Overall, the technical contribution of the work look sound!
The theorectical aspect of the evolving rate is valuable in modeling sequential data!

**Weaknesses:**

The experimental details about the cost function of the optimal transport in aligning data across time steps, the parameter and the neural architecture, are not provided. These details could be crucial for the others to use the methods!

**Questions:**

How to choose the cost function in using optimal transport to build correspondence for data samples across time steps? Can you provide more experimental details?

**Limitations:**

No, the authors do not discuss the potential negative societal impact of their work.

---

> ### Author Rebuttal · Authors · 2024-08-07
>
> Responses to Weakness
> 1. (Cost function) The cost function of Optimal Transport (OT) is defined in Eq (7). In the literature [1-2], OT typically uses a distance metric between two samples, each sampled from different marginal distributions, as the cost function. For instance, $z^s$ and $z^t$ represent data from source and target domains in domain adaptation tasks [1-2]. In our approach, we account for the dynamic characteristics of sequential data. Specifically, we assume the presence of an autoregressive function $f$ that governs the evolving patterns within the data. Consequently, we design the cost function as the distance between the output of $f$ with historical samples as input $f(\\textbf{z}^t_{t-k+1})$, and the subsequent sample $z_{t+1}$. In our implementation, OT aims to find the matching between a batch of samples $\\{z_{t,i}\\}^B_{i=1}$ and $\\{z_{t+1,i}\\}^B_{i=1}$ in a training iteration across time steps.
>
> 2.  (parameter and architecture) Thanks for pointing it out. We will add this detail in the revision. The autoregressive model $f$ is typically a 2 or 3-layer LSTM, with input size, hidden size, and output size set to 128, 256, or 512, depending on the architecture of the encoder $g$. For tabular data, $g$ is a 2 or 3-layer MLP, with the input size corresponding to the data size, and the hidden and output sizes set to either 128 or 512. For image data, $f$ is a ResNet-12 with a hidden size of 512.
>
>
> Q: (choose the cost function) We do not select the cost function; rather, we use a cost function that is consistent with the critic function as an autoregressive form in mutual information estimation, which leverages the structure of sequential data.
>
> [1] Optimal Transport for Domain Adaptation, TPAMI 2015
>
> [2] Joint distribution optimal transportation for domain adaptation, NIPS 2017

---

### Official Review · Reviewer_oZmm · 2024-07-12

**Soundness:** 4
**Presentation:** 4
**Contribution:** 3
**Rating:** 8
**Confidence:** 4

**Summary:**

The article introduces a groundbreaking technique for measuring changes in sequential data, marking a notable advancement in the realm of machine learning. The authors bring forth the Evolving Rate (EvoRate) and its advanced iteration, EvoRate$_\mathcal{W}$, as effective tools for evaluating the temporal dynamics within datasets.

**Strengths:**

- Tackles a real-world issue in machine learning, with possible uses in predicting time-series data, categorization, and other sectors.
- Bridges a knowledge gap in the current comprehension and measurement of patterns that change over time in sequential data.
- The creation of EvoRate and EvoRate$_\mathcal{W}$ represents a creative leap forward, offering a numerical assessment of evolving patterns applicable to a range of learning tasks. The utilization of optimal transport (OT) to resolve the issue of non-correspondence in temporal data is especially innovative.

**Weaknesses:**

- It may not fully address the scalability of the proposed methods as dimensionality grows. The authors assert that the dimensionality reduction of the original data can lead to a decrease in computational time when calculating Mutual Information (MI). However, the process of training an auto-encoder itself demands considerable time and resources. Is this factored into their considerations?
- The scope of the experiments might be limited, with a notable absence of extensive real-world datasets
- The true MI values for high-dimensional data, such as in video and time series contexts, are often infeasible to obtain. How do the authors ensure that EvoRate can provide an accurate measure of MI for such high-dimensional data, especially given this significant limitation?

**Questions:**

-I find the concept of leveraging Mutual Information (MI) to evaluate the evolution of patterns to be logical, yet I harbor reservations about whether MI can truly equate to the changes observed in sequential data. A deeper exploration of this connection would be highly beneficial.

-The ForeCA method's results are counterintuitive, as they oppose the prediction performance outlined in Table 1, where one would expect the inclusion of more historical data to enhance the predictive capabilities. Can the authors elucidate why EvoRate, an estimator of MI, is deemed superior for capturing evolving patterns over ForeCA, which is based on entropy estimation?

-I am curious to know if the EvoRate model, once trained, can be directly applied to new datasets without the need for retraining. If this is possible, it would make EvoRate a more adaptable and computationally efficient tool that could be easily integrated into various systems.

**Limitations:**

There's no limitation relating to social impact.

---

> ### Author Rebuttal · Authors · 2024-08-07
>
> Responses to Weakness:
>
> 1. (scalability & training an auto-encoder) We have added experiments on the scalability of the method with different dimensions in the encoding space using the Video Prediction dataset KITTI.
>
>     Encoding Dim|128| 256 |512| 1024
>     |-------------------------|-------------------------------|-----------------------------|-------------------------------|-----------------------------|
>     |EvoRate|2.19| 2.37|2.47| 2.56|
>
>     From the above table, a smaller dimension in the encoding space leads to a lower EvoRate due to greater information loss. However, the performance remains acceptable even with a very low dimension of 128.
>
>     Training an auto-encoder model from scratch can be computationally expensive. However, we can use a pretrained backbone, significantly reducing computational costs by directly learning the dynamic function $f$ from the encoded embeddings.
>
>     |     Corruption Ratio           |0                       |0.05|0.1|0.25|0.5                         |
>     |-------------------------|-------------------------------|-----------------------------|-------------------------------|-----------------------------|----------------------------|
>     |Finetune $g$ | 2.56|1.52|0.93|0.37|0.05
>     |Pretrained&fixed $g$ |2.54| 1.55| 0.92|0.39|0.10
>
>     The Corruption Ratio represents the probability of shuffling the sequence to disrupt the evolving patterns. A higher Corruption Ratio should result in a lower EvoRate. We find that with a fixed pretrained encoding function $g$, the estimated EvoRate closely approximates the result of training $g$ with EvoRate on the video dataset KITTI. The pretrained model used is ResNet-18 from torchvision, pretrained on ImageNet-1K.
>
>
> 2. (Absence of real-world datasets) We have utilized real-world datasets: five time-series forecasting datasets presented in Table 2, three EDG datasets (Portraits, Caltran, and Powersupply) in Table 3, and one video prediction dataset (KITTI) in Figure 2-c. Additionally, we included the experimental result of EvoRate on the NLP dataset Associative Recall from zoology [3]. A low EvoRate for the NLP dataset implies either a low prediction potential or that this dataset alone cannot be used to train a good predictive model. Notably, the test accuracy of AR is typically 100%, which is consistent with the high EvoRate of 2.93.
>
>     |     Corruption Ratio           |0                       |0.25|0.5|0.75|1                         |
>     |----------------|-------------------------------|-----------------------------|-------------------------------|-----------------------------|----------------------------|
>     |EvoRate |2.93| 1.52|1.28|1.18|1.04|
>
>
> 3. (True MI are not able to obtain) We acknowledge that for real-world high-dimensional data, MI is often intractable, as highlighted in the literature [1]. However, our experimental results demonstrate that we can achieve a reasonable estimation of MI, as illustrated in Figure 1. Similar to the approach of ForeCA, we use an information-theory-based estimator as an indicator of evolving patterns. While ForeCA uses entropy, we use a mutual information estimator.
>
>
>
> Responses to Questions:
>
> 1. (MI to evaluate evolution of patterns) Using MI to estimate the relationship between pairs of variables is a fundamental problem in science and engineering. We propose to leverage MI to quantify the strength of the evolving patterns, which reveals that the evolving patterns is strongly related to the latent temporal dependency. We justify our choice of using MI to measure evolving patterns in Section 4.2 and Proposition 1. Specifically, MI is an intrinsic property of the data that directly influences the expected error of the maximum likelihood estimation loss. Therefore, we claim that MI serves as a direct and reliable indicator of evolving patterns.
>
> 2. (Why EvoRate better than ForeCA) This limitation is due to ForeCA's inability to detect complex patterns, as illustrated in lines 112-114 of our paper. While temporal patterns can encompass trends, cycles, irregular fluctuations, and more complex behaviors, ForeCA is restricted to detecting only simple linear cyclical patterns. In his paper [2], he assumes that the data from future time steps is a linear transformation of the historical data. However, our methods can leverage MI-guided efficient deep representation learning to describe complex and non-linear evolving patterns.
>
> 3. (apply to new datasets without finetuning) We acknowledge that the current EvoRate model can utilize a fixed pre-trained encoder $g$ to project data into a lower dimension to reduce computational complexity. However, the autoregressive function $f$ in Eq (4) and (9) still requires fine-tuning for each dataset due to the distinct evolving patterns present in different datasets. Nonetheless, training a larger EvoRate model with more datasets could potentially result in a more robust model capable of testing evolving patterns across various datasets. This is a promising direction for our future research.
>
> [1] Estimating mutual information. Physical Review 2004
>
> [2] Forecastable component analysis. ICML 2013
>
> [3] Zoology: Measuring and Improving Recall in Efficient Language Models, Arxiv 2023

---

> > ### Comment · Reviewer_oZmm · 2024-08-10
> >
> > Thank you for the response. I've read it carefully and my questions have been addressed. I will keep my current score.

---

> > > ### Author Response · Authors · 2024-08-10
> > >
> > > Dear Reviewer oZmm,
> > >
> > > Thank you for your kind response and for taking the time to review our responses. We are pleased to hear that our responses were satisfactory and that you are maintaining your original assessment. Your thoughtful feedback has been invaluable in strengthening our work, and we sincerely appreciate your efforts.
> > >
> > > Best regards,
> > > Authors

---

### Official Review · Reviewer_rYZr · 2024-07-12

**Soundness:** 4
**Presentation:** 3
**Contribution:** 4
**Rating:** 8
**Confidence:** 5

**Summary:**

This paper introduces EvoRate, a novel metric designed to quantify the evolving patterns in sequential data. The authors propose leveraging mutual information (MI) to measure the temporal dependencies between data points in a sequence. The paper addresses a significant challenge in machine learning: identifying and quantifying evolving patterns in high-dimensional sequential data. EvoRate is further extended to EvoRate$_W$, which uses optimal transport to establish correspondences between data points at different timestamps, facilitating MI estimation even when direct correspondences are absent. The proposed methods are validated through experiments on both synthetic and real-world datasets, demonstrating their effectiveness in various applications such as time-series forecasting, classification with temporal distribution shifts, and video prediction.

**Strengths:**

1. It is true that most researchers apply sequential models to sequential data without considering whether there are evolving patterns. Therefore, it is critical to delve deeper into understanding these evolving patterns in sequential data. The introduction of EvoRate and its extension, EvoRate$_W$, addresses a significant gap in the literature by providing a quantitative measure for evolving patterns in sequential data. This is a novel and impactful contribution.

2. The paper provides analysis and examples to clarify why EvoRate can measure evolving patterns, particularly through Proposition 1. This proposition reveals that mutual information (MI) is an intrinsic property of sequential data, reflecting the evolving patterns within the data. Detailed explanations of how MI quantifies temporal dependencies and the use of optimal transport for handling data without direct correspondences are included, enhancing the understanding of this measure.

3. Figure 1 (a, b) illustrates the designed MI critic function used in Equation 4, which better measures MI by considering the data structure of sequential data. More importantly, Figure 1 (c, d) demonstrates the effectiveness of EvoRate$_W$. Even in the absence of correspondence between time points, EvoRate$_W$ can still approximate the ground truth MI, showcasing its robustness and practical utility.

**Weaknesses:**

1. The authors refer to EDG as their algorithmic contribution in Line 70, but the detailed setting and definition of the EDG problem are lacking. A clear introduction and definition are necessary to understand the context and setup of EDG.

2. There is some confusion regarding the term "learning performance" mentioned in Line 53. It is unclear whether this refers to the prediction tasks of sequential data or the process of training the model to measure mutual information (MI). Additionally, how EvoRate manages the trade-off between computational complexity and learning performance needs clarification. Since the mutual information measurement model also requires training, as is standard in variational MI estimators [1], why is it necessary to measure MI to reflect evolving patterns rather than directly relying on the predictive performance of sequential models (e.g., LSTM, Attention, SSM), which also require training?

3. It would be beneficial to include a pseudocode table to facilitate a clearer understanding of the algorithms for the readers.

4. Although the authors state they do not have experiments on NLP tasks, it would be highly recommended to apply EvoRate to NLP datasets, given that NLP is one of the most important sequential data tasks.

5. The claim that the function $f$ will converge to $f^*$ in Remark 2 is unconvincing. This is because the joint distribution is not available, making it difficult to accurately learn the transition function $f$. Without access to the joint distribution, the learning process lacks the necessary information to ensure convergence to the optimal function $f^*$, especially when dealing with complex, high-dimensional real-world data.

6. Synthetic experiments with known correspondences in Section 6.1, only include the average operation as the transition function. It would be more beneficial to also consider a more practical dataset.

7. There are some typos that cause confusions: Table 2 should be EvoRate and Table 3 should be EvoRate$_W$.

[1] Mohamed Ishmael Belghazi, Aristide Baratin, Sai Rajeshwar, Sherjil Ozair, Yoshua Bengio, Aaron Courville, and Devon Hjelm. Mutual information neural estimation. In Jennifer Dyand Andreas Krause, editors, Proceedings of the 35th International Conference on Machine Learning, volume 80 of Proceedings of Machine Learning Research, pages 531–540. PMLR, 10–15 Jul 2018.

**Questions:**

See the weaknesses above.

**Limitations:**

This paper is about fundamental machine learning research and is unlikely to have any potential negative social impact.

---

> ### Author Rebuttal · Authors · 2024-08-07
>
> 1. (EDG setting) We thank you for pointing it out and will include an illustration of the Evolving Domain Generalization (EDG) setup. A brief explanation of EDG can be found in lines 328-330. More specifically, The EDG tasks setup involves using the training dataset $D_S = \\{ \\{ x_{t,i}, y_{i,t} \\}^N_{i=1} \\}^M_{t=1}$, where $x_{i,t}$ and $y_{i,t}$ represent the data and its label of the $i$-th sample at time $t$ respectively and $N$ is the sample size, to learn a function that can predict the class of samples from future timestamps, which are then evaluated on the test dataset $D_T = \\{ \\{ x_{i,M+t}, y_{i,M+t} \\}^N_{i=1}\\}^L_{t=1}$.
>
>
> 2. - (a: learning performance) Thanks for pointing it out. The term "learning performance" refers to the performance of approximating MI using the trained EvoRate, which is parameterized by the learned autoregressive function $f$ and the encoder function $g$.
>
>     - (b: trade-off between complexity and performance) When the original dimension of the data $D$ equals the dimension of the encoded data's embedding $d$, it provides the most precise MI estimation, comparable to the computational cost of directly training a deep autoregressive model. On the other hand, when $d \ll D$, the estimated precision is lower but more computationally efficient. Specifically, if we directly load a pretrained model for $g$, we only need to train $f$, which is typically only composed of multi-layer perceptrons (MLPs).
>
>     - (c: why measure MI instead of relying on the predictive performance) As pointed out in (b), setting $d \ll D$ makes EvoRate efficient in estimating MI compared to training a deep autoregressive model in the original high dimension, especially for high-resolution videos. The key is to estimate MI in the latent encoding space. We can trade off some precision of EvoRate due to the Data Processing Inequality [1] for computational cost efficiency. We provide additional experimental results with the KITTI dataset:
>
>         |     Corruption Ratio           |0                       |0.05|0.1|0.25|0.5                         |
>         |-------------------------|-------------------------------|-----------------------------|-------------------------------|-----------------------------|----------------------------|
>         |Finetune $g$ | 2.56|1.52|0.93|0.37|0.05
>         |Pretrained&fixed $g$ |2.54| 1.55| 0.92|0.39|0.10
>
>         In the above table, the Corruption ratio represents the probability of shuffling the data sequence, where a high ratio leads to a degradation of evolving patterns.
>
>
> 3. (pseudocode) We have included the pseudocode in the global response PDF and will add it to the revised version of the paper.
>
> 4. (NLP tasks) We further experiment with EvoRate on the NLP dataset Associative Recall (AR) in zoology [4], and the results are shown below. Due to time limit, we choose this simple synthetic dataset for our experiments. EvoRate is estimated between the historical sequence and its query's recall. A low EvoRate for the NLP dataset implies either a low prediction potential or that this dataset alone cannot be used to train a good predictive model. Notably, the test accuracy of AR is typically 100%, which is consistent with the high EvoRate of 2.93, indicating strong predictive performance..
>
>
>     |     Corruption Ratio           |0                       |0.25|0.5|0.75|1                         |
>     |----------------|-------------------------------|-----------------------------|-------------------------------|-----------------------------|----------------------------|
>     |EvoRate |2.93| 1.52|1.28|1.18|1.04|
>
> 5. ($f$ will converge to $f^*$) In Lemma 1, we show that with our defined autoregressive cost function, if $f$ attains $f^*$, the estimated joint distribution $\pi^*$ will converge to the real distribution. Furthermore, it is a common practice to estimate the joint distribution with optimal transport to build the correspondence, as demonstrated in [2,3]. However, proving the convergence of $f$ to $f^*$ is challenging, and we will try to address this in future research.
>
> 6. (Synthetic experiments) For more complex transition functions, the analytical mutual information value is intractable, as there are no methods to compute the ground truth MI value quantitatively. For datasets with correspondences, we test EvoRate on real-world time series forecasting datasets Crypto, Player Traj., M4-Monthly, M4-Weekly, M4-Daily, and video prediction KITTI dataset.
>
> 7. (typos) Thank you for pointing them out, and we will revise the paper accordingly.
>
> [1] Elements of information theory. John Wiley & Sons, 1999
>
> [2] Joint distribution optimal transportation for domain adaptation, NIPS 2017
>
> [3] Optimal Transport for Domain Adaptation, TPAMI 2015
>
> [4] Zoology: Measuring and Improving Recall in Efficient Language Models, Arxiv 2023

---

> > ### Comment · Reviewer_rYZr · 2024-08-12
> >
> > Thank you for the clarifications and the new experiments. After reviewing the rebuttal, my concerns have been fully addressed. I am particularly impressed that EvoRate can be effectively applied to NLP datasets, demonstrating significant potential for various real-world applications. I have no further questions and am pleased to raise my score.

---

> > > ### Author Response · Authors · 2024-08-12
> > >
> > > Dear Reviewer  rYZr,
> > >
> > > We greatly appreciate your feedback and are glad that our rebuttal has resolved your concerns, resulting in an improved rating! We will incorporate your suggestions into the manuscript in future revisions.
> > >
> > > Best regards,
> > >
> > > Authors of 6086

---

### Official Review · Reviewer_8iST · 2024-07-13

**Soundness:** 2
**Presentation:** 2
**Contribution:** 3
**Rating:** 6
**Confidence:** 4

**Summary:**

This paper aims to identify the evolving pattern in sequential data. In addition, given evolving pattern may present in the sequential data, this paper would like to introduce a technique that can identify the best temporal order and features for learning the sequential data.
To address that, this paper proposed an indicator Evolving Rate (EvoRate), by borrowing the ideas from existing works in Mutual Information (MI) Estimation. The effectiveness of EvoRate is tested by experiments on multivariate time series forecasting, video forecasting and evolving Domain Generalization (EDG) tasks.

**Strengths:**

The research question discussed is interesting.

**Weaknesses:**

This paper measures evolving patterns by an improved MI indicator. Some discussions and experiments are conducted to evaluate the advantages of EvoRate over existing MIs. However, the question discussed in this paper is "How to determine the existence of evolving patterns in data sequences" which does not limit the technical solutions to MI-based one. Therefore, I am questioning the contribution of this paper.

**Questions:**

Apart from comparing to existing works in MI, do you also consider studies working on evolving patterns but not using MI? How to use the results of accurate identification of evolving patterns? This paper shows the results of the estimated EvoRate compared to MI results. However, it is less clear to me what the real benefit of introducing MI into solving the evolving pattern issue.

**Limitations:**

This paper shows some interesting ideas and results, but the motivation is not very clear to me. The proposed method is an edited version of MI. However, I am not clear why it is beneficial to introduce MI to address evolving patterns compared to existing solutions.

---

> ### Author Rebuttal · Authors · 2024-08-07
>
> - W1 & Limitation (motivation is not very clear) - **Our motivations and contributions**: Our work's contribution is three-fold:
>   - It is theoretically motivated by the use of MI to estimate evolving patterns.
>   - It applies a specially designed similarity critic that considers the autoregressive manner.
>   - For cases where data is sampled at each timestamp and lacks correspondence, we propose methods to approximate the absent correspondence between timestamps with optimal transport (OT), thus enabling EvoRate estimation.
>
> - W1 & Q1 (benefit of introducing MI) - **Why we use MI**: We justify our choice of using MI in Section 4.2 and Proposition 1. Specifically, we show that **a larger MI leads to a smaller expected maximum likelihood estimation (MLE) loss, which indicates better autoregression performance**. As shown in Eq (5), the expected risk can be decomposed into two terms: the first term is related to the prediction model, and the second term, $H(Z_{t+1}) - I(Z_{t+1};\textbf{Z}^t_{t-k+1})$, is only related to the data. Since the inequality $H(Z_{t+1}) \ge I(Z_{t+1}; \textbf{Z}^t_{t-k+1})$ always holds, minimizing the prediction risk in autoregressions requires MI to be as close as possible to $H(Z_{t+1})$. The insight behind achieving equality is that we can predict the future state $Z_{t+1}$ if we have complete information about the historical states $\textbf{Z}_{t-k+1}^t$. Consequently, we argue that MI serves as a direct and reliable indicator of evolving patterns. Based on this conclusion, our experiments aim to show that EvoRate can effectively approximate MI in sequential data.
>
> - W1 & Q1 (comparing to existing works) - **Advantages over the existing work**: The problem of evolving patterns in sequential data is under-studied, and to the best of our knowledge, the only related work is ForeCA [1]. However, ForeCA relies on the spectral entropy of data and assumes linear dependency between historical information and future states. Consequently, it can only detect the cyclical pattern and cannot serve as an indicator of complex evolving patterns. Moreover, ForeCA cannot be applied to high-dimensional data, limiting its use for many real-world datasets. EvoRate addresses these issues by MI-guided efficient deep representation learning, and its effectiveness is justified in Table 1 of our paper. More discussions of the benefits of EvoRate over ForeCA are also presented in lines 106-116 of our paper. We also include three statistic indicators for time series in global response.
>
> - Q1 (how to use results of identification of evolving patterns): There are various applications:
>     - a) Model selection (sequential vs. static): By establishing an empirical threshold for EvoRate, we can determine the appropriate model for predictions. If EvoRate exceeds the threshold, a sequential model is recommended. Conversely, if EvoRate is below the threshold, indicating a lack of temporal patterns (e.g., coin tossing), a static model should be used. For example, based on the results in Table 3, we should apply a static model for the Portraits and Caltran datasets.
>     - b) Temporal order estimation: EvoRate can be used to estimate the temporal order in the data. From Table 1, we suggest building a temporal model with a temporal order of 90 can achieve optimal performance, as EvoRate values for orders 180 and 270 are very close to that of 90.
>     - c) Temporal feature selections: EvoRate can be used for feature selection for temporal prediction tasks, allowing for efficient and explainable model training by excluding redundant or irrelevant features, as shown in Figure 2-b.
>     - d) Evolving domain generalization (EDG): EvoRate naturally acts as a regularizer to enhance the performance of learning dynamics in sequential data, as verified in EDG tasks in Table 4.
>     - e) We further experiment with EvoRate on the NLP dataset Associative Recall (AR) of zoology [4]. The results indicate that a low EvoRate for the NLP dataset implies either a low prediction potential or that the dataset alone cannot be used to train a good predictive model. EvoRate is estimated between the historical sequence and its query's recall.
>
>         |Corruption Ratio|0   |0.25|0.5|0.75|1|
>         |-|-|-|-|-|-|
>         |EvoRate |2.93| 1.52|1.28|1.18|1.04|
>
>         The Corruption ratio represents the probability of shuffling the data sequence. A high ratio leads to a degradation of evolving patterns, resulting in a lower EvoRate. This is consistent with the results shown in the above table. Notably, the test accuracy of the AR dataset is normally 100%, which aligns with its high EvoRate of 2.93.
>
> - Limitation (EvoRate is edited MI): There are two major differences between EvoRate and existing MI estimation methods (e.g., [5-6]): 1) Existing MI estimation methods are designed for static data, and their concatenated and separable critic functions directly compute the similarity between embeddings in two different domains. In contrast, EvoRate leverages an autoregressive critic function to efficiently account for the structure of sequential data by mapping historical states to the next state and computing similarity in the domain of the next state. As a result, EvoRate underestimates the ground truth MI values of sequential data, as shown in Figure 1-b. 2) Existing MI estimation methods do not account for sequential data without correspondences, and hence cannot directly estimate MI in such cases. EvoRate$_W$ mitigates this issue by building correspondence with optimal transport (OT). However, even with OT, existing MI estimation methods still fail to estimate MI for sequential data due to limitations in their critic functions, as shown in Figure 1.
>
> [1] Forecastable component analysis
>
> [2] Lyapunov exponents
>
> [3] Testing stationarity in time series
>
> [4] Measuring and Improving Recall in Efficient Language Models
>
> [5] Mutual information neural estimation
>
> [6] A contrastive log-ratio upper bound of mutual information

---

> > ### Comment · Reviewer_8iST · 2024-08-14
> >
> > Thanks for good explanation for my concerns

---

> > > ### Author Response · Authors · 2024-08-14
> > >
> > > Thank you for raising our score. We highly appreciate your recognition of our work.

---

> > ### Author Response · Authors · 2024-08-14
> >
> > We have identified a typo in our rebuttal:
> >
> > In our response to the Limitation section, the phrase "As a result, EvoRate underestimates..." should instead refer to existing MI methods, not EvoRate.

---

### Author Rebuttal · Authors · 2024-08-07

We would like to thank all the reviewers for their valuable comments on our work.

We have received four reviews with ratings of 3, 6, 7, and 7.

We are pleased that the reviewers have good impressions of our work, including:

- Addressing an interesting and important problem (8iST, rYZr, V8vA);
- Presenting a well-motivated paper with the proposed method EvoRate being useful for a variety of applications (rYZr, V8vA, oZmm);
- Theoretical justification for choosing mutual information (MI) as the indicator of evolving patterns in sequential data (rYZr, V8vA);
- Generally clear presentation (V8vA).

During the rebuttal period, we have provided detailed responses to all comments and questions point by point. Specifically,

- We further clarified why we chose Mutual Information as the metric to measure evolving patterns (to 8iST & oZmm).
- We further clarified how we trade off computational cost with the performance of EvoRate by selecting the dimension of the encoding space and added experiments to address this with different dimensions of the encoding space (to rYZr & oZmm).
- We added an NLP dataset to evaluate EvoRate, demonstrating its capability in real-world applications (to rYZr & oZmm).
- We added experiments with a fixed pretrained encoder to show that EvoRate can be very efficient by only training the autoregressive model $f$ (to oZmm).

In the attached PDF, we provide the pseudocode for our proposed algorithms, a detailed comparison of three traditional time-series statistical indicators with EvoRate, and an introduction to the newly added NLP dataset.

Lastly, we would like to thank all the reviewers for their time once again. Could you please check our response and confirm if you have any further questions? **We are looking forward to your post-rebuttal feedback!**

---

### Decision · Program_Chairs · 2024-09-25

**Decision:**

Accept (spotlight)

**Comment:**

The paper examines the underlying assumptions made by autoregressive time series models that data streams are evolving, by introducing a method EvoRate that seeks to detect whether a data stream is evolving or not by looking at mutual information across the stream. They introduce a method EvoRateW to detect EvoRate using optimal transport.

After the author response period, the paper has scores of 6, 8, 8 and 6. The reviewers agreed that the method has potential for real world applicability, with one reviewer particularly impressed by the applicability to NLP. The reviews also agreed that there is significant novelty in the methodology.

I recommend this paper for acceptance, with the suggested changes being included.

Minor comment - I noticed that citation [30] has formatting errors - I would recommend a thorough proof-reading before .camera ready submission.